# VQ-GNN: A Universal Framework to Scale-up Graph Neural Networks using Vector Quantization

**Mucong Ding**[*], **Kezhi Kong**[*], **Jingling Li** , **Chen Zhu** ,
**John Dickerson** , **Furong Huang** , **Tom Goldstein**

Department of Computer Science, University of Maryland
*{mcding, kong, jingling, chenzhu, john, furongh, tomg}@cs.umd.edu*

## Abstract

Most state-of-the-art Graph Neural Networks (GNNs) can be defined as a form of graph convolution which can be realized by message passing between direct neighbors or beyond. To scale such GNNs to large graphs, various neighbor-, layer-, or subgraph-sampling techniques are proposed to alleviate the "neighbor explosion" problem by considering only a small subset of messages passed to the nodes in a mini-batch. However, sampling-based methods are difficult to apply to GNNs that utilize many-hops-away or global context each layer, show unstable performance for different tasks and datasets, and do not speed up model inference. We propose a principled and fundamentally different approach, VQ-GNN, a universal framework to scale up any convolution-based GNNs using Vector Quantization (VQ) without compromising the performance. In contrast to sampling-based techniques, our approach can effectively preserve all the messages passed to a mini-batch of nodes by learning and updating a small number of quantized reference vectors of global node representations, using VQ within each GNN layer. Our framework avoids the "neighbor explosion" problem of GNNs using quantized representations combined with a low-rank version of the graph convolution matrix. We show that such a compact low-rank version of the gigantic convolution matrix is sufficient both theoretically and experimentally. In company with VQ, we design a novel approximated message passing algorithm and a nontrivial back-propagation rule for our framework. Experiments on various types of GNN backbones demonstrate the scalability and competitive performance of our framework on large-graph node classification and link prediction benchmarks.

## 1 Introduction

The rise of Graph Neural Networks (GNNs) has brought the modeling of complex graph data into a new era. Using message-passing, GNNs iteratively share information between neighbors in a graph to make predictions of node labels, edge labels, or graph-level properties. A number of powerful GNN architectures [1–4] have been widely applied to solve down-stream tasks such as recommendation, social analysis, visual recognition, etc.

With the soaring size of realistic graph datasets and the industrial need to model them efficiently, GNNs are hindered by a scalability problem. An $L$-layer GNN aggregates information from all $L$-hop neighbors, and standard training routines require these neighbors to all lie on the GPU at once. This prohibits full-batch training when facing a graph with millions of nodes [5].

---

[*]Equal contribution.

35th Conference on Neural Information Processing Systems (NeurIPS 2021).

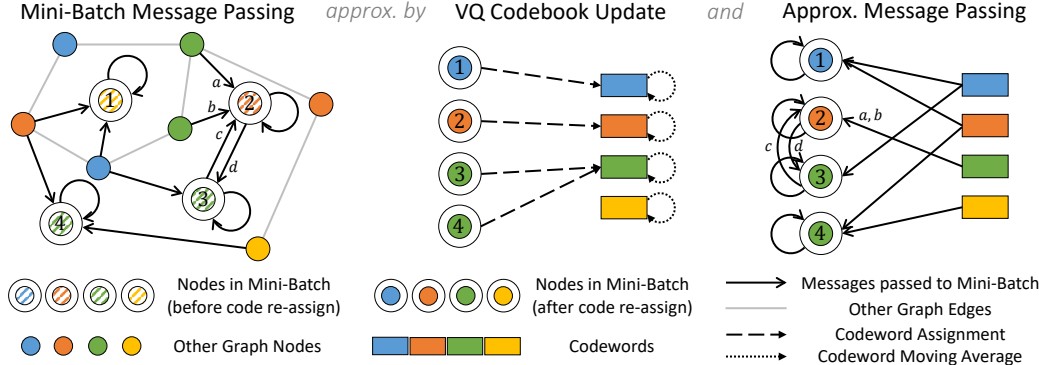

Figure 1: In our framework, VQ-GNN, each *mini-batch message passing* (left) is approximated by a *VQ codebook update* (middle) and an *approximated message passing* (right). All the messages passed to the nodes in the current mini-batch are effectively preserved. Circles are nodes, and rectangles are VQ codewords. A double circle indicates nodes in the current mini-batch. Color represents codeword assignment. During *VQ codebook update*, codeword assignment of nodes in the mini-batch is refreshed (node 1), and codewords are updated using the assigned nodes. During *approximated message passing*, messages from out-of-mini-batch nodes are approximated by messages from the corresponding codewords, messages from nodes assigned to the same codeword are merged ($a$ and $b$), and intra-mini-batch messages are not changed ($c$ and $d$).

A number of sampling-based methods have been proposed to accommodate large graphs with limited GPU resources. These techniques can be broadly classified into three categories: (1) Neighbor-sampling methods [2, 6] sample a fixed-number of neighbors for each node; (2) Layer-sampling methods [7, 8] sample nodes in each layer independently with a constant sample size; (3) Subgraph-sampling methods [9, 10] sample a subgraph for each mini-batch and perform forward and back-propagation on the same subgraph across all layers. Although these sampling-based methods may significantly speed up the training time of GNNs, they suffer from the following three major draw-backs: (1) At inference phase, sampling methods require all the neighbors to draw non-stochastic predictions, resulting in expensive predictions if the full graph cannot be fit on the inference device; (2) As reported in [5] and in Section 6, state-of-the-art sampling-baselines fail to achieve satisfactory results consistently across various tasks and datasets; (3) Sampling-based methods cannot be universally applied to GNNs that utilize many-hop or global context in each layer, which hinders the application of more powerful GNNs to large graphs.

This paper presents VQ-GNN, a GNN framework using vector quantization to scale most state-of-the-art GNNs to large graphs through a principled and fundamentally different approach compared with the sampling-based methods. We explore the idea of using vector quantization (VQ) as a means of dimensionality reduction to learn and update a small number of quantized reference vectors (codewords) of global node representations. In VQ-GNN, mini-batch message passing in each GNN layer is approximated by a VQ codebook update and an approximated form of message passing between the mini-batch of nodes and codewords; see Fig. 1. Our approach avoids the "neighbor explosion" problem and enables mini-batch training and inference of GNNs. In contrast to sampling-based techniques, VQ-GNN can effectively preserve all the messages passed to a mini-batch of nodes. We theoretically and experimentally show that our approach is efficient in terms of memory usage, training/inference time, and convergence speed. Experiments on various GNN backbones demonstrate the competitive performance of our framework compared with the full-graph training baseline and sampling-based scalable algorithms.

**Paper organization.**   The remainder of this paper is organized as follows. Section 2 summarizes GNNs that can be re-formulated into a common framework of graph convolution. Section 3 defines the scalability challenge of GNNs and shows that dimensionality reduction is a potential solution. In Section 4, we describe our approach, VQ-GNN, from theoretical framework to algorithm design and explain why it solves the scalability issue of most GNNs. Section 5 compares our approach to the sampling-based methods. Section 6 presents a series of experiments that validate the efficiency, robustness, and universality of VQ-GNN. Finally, Section 7 concludes this paper with a summary of limitations and broader impacts.

Table 1: Summary of GNNs re-formulated as generalized graph convolution.

| Model Name | Design Idea | Conv. Matrix Type | # of Conv. | Convolution Matrix |
|---|---|---|---|---|
| GCN[1] [1] | Spatial Conv. | Fixed | 1 | $C = \widetilde{D}^{-1/2}\widetilde{A}\widetilde{D}^{-1/2}$ |
| SAGE-Mean[2] [2] | Message Passing | Fixed | 2 | $\begin{cases} C^{(1)} = I_n \\ C^{(2)} = D^{-1}A \end{cases}$ |
| GAT[3] [3] | Self-Attention | Learnable | # of heads | $\begin{cases} \mathfrak{C}^{(s)} = A + I_n \text{ and} \\ h_{\boldsymbol{a}^{(l,s)}}^{(s)}(X_{i,:}^{(l)}, X_{j,:}^{(l)}) = \exp\big(\text{LeakyReLU}( \\ \quad (X_{i,:}^{(l)}W^{(l,s)} \,\|\, X_{j,:}^{(l)}W^{(l,s)}) \cdot \boldsymbol{a}^{(l,s)})\big) \end{cases}$ |

[1] Where $\widetilde{A} = A + I_n$, $\widetilde{D} = D + I_n$.    [2] $C^{(2)}$ represents mean aggregator. Weight matrix in [2] is $W^{(l)} = W^{(l,1)} \,\|\, W^{(l,2)}$.
[3] Need row-wise normalization. $C_{i,j}^{(l,s)}$ is non-zero if and only if $A_{i,j} = 1$, thus GAT follows direct-neighbor aggregation.

## 2 Preliminaries: GNNs defined as Graph Convolution

**Notations.** Consider a graph with $n$ nodes and $m$ edges (average degree $d = m/n$). Connectivity is given by the adjacency matrix $A \in \{0,1\}^{n \times n}$ and features are defined on nodes by $X \in \mathbb{R}^{n \times f_0}$ with $f_0$ the length of feature vectors. Given a matrix $C$, let $C_{i,j}$, $C_{i,:}$, and $C_{:,j}$ denote its $(i,j)$-th entry, $i$-th row, $j$-th column, respectively. For a finite sequence $\langle i_b \rangle : i_1, \dots, i_b$, we use $C_{\langle i_b \rangle,:}$ to denote the matrix whose rows are the $i_b$-th rows of matrix $C$. We use $\odot$ to denote the element-wise (Hadamard) product. $\|\cdot\|_p$ denotes the entry-wise $\ell^p$ norm of a vector and $\|\cdot\|_F$ denotes the Frobenius norm. We use $I_n \in \mathbb{R}^{n \times n}$ to denote the identity matrix, $\mathbf{1}_n \in \mathbb{R}^n$ to denote the vector whose entries are all ones, and $\boldsymbol{e}_n^i$ to denote the unit vector in $\mathbb{R}^n$ whose $i$-th entry is 1. The 0-1 indicator function is $\mathbb{1}\{\cdot\}$. We use $\text{diag}(\boldsymbol{c})$ to denote a diagonal matrix whose diagonal entries are from vector $\boldsymbol{c}$. And $\|$ represents concatenation along the last axis. We use superscripts to refer to different copies of same kind of variable. For example, $X^{(l)} \in \mathbb{R}^{n \times f_l}$ denotes node representations on layer $l$. A Graph Neural Network (GNN) layer takes the node representation of a previous layer $X^{(l)}$ as input and produces a new representation $X^{(l+1)}$, where $X = X^{(0)}$ is the input features.

**A common framework for generalized graph convolution.** Although many GNNs are designed following different guiding principles including neighborhood aggregation (GraphSAGE [2], PNA [11]), spatial convolution (GCN [1]), spectral filtering (ChebNet [12], CayleyNet [13], ARMA [14]), self-attention (GAT [3], Graph Transformers [15–17]), diffusion (GDC [18], DCNN [19]), Weisfeiler-Lehman (WL) alignment (GIN [4], 3WL-GNNs [20, 21]), or other graph algorithms ([22, 23]). Despite these differences, *nearly all GNNs can be interpreted as performing message passing on node features, followed by feature transformation and an activation function.* As pointed out by Balcilar et al. [24], GNNs can typically be written in the form

$$X^{(l+1)} = \sigma\left(\sum_s C^{(s)} X^{(l)} W^{(l,s)}\right), \tag{1}$$

where $C^{(s)} \in \mathbb{R}^{n \times n}$ denotes the $s$-th convolution matrix that defines the message passing operator, $s \in \mathbb{Z}_+$ denotes index of convolution, and $\sigma(\cdot)$ denotes the non-linearity. $W^{(l,s)} \in \mathbb{R}^{f_l \times f_{l+1}}$ is the learnable linear weight matrix for the $l$-th layer and $s$-th filter.

Within this common framework, GNNs differ from each other by choice of convolution matrices $C^{(s)}$, which can be either fixed or learnable. A learnable convolution matrix relies on the inputs and learnable parameters and can be different in each layer (thus denoted as $C^{(l,s)}$):

$$C_{i,j}^{(l,s)} = \underbrace{\mathfrak{C}_{i,j}^{(s)}}_{\text{fixed}} \cdot \underbrace{h_{\theta^{(l,s)}}^{(s)}(X_{i,:}^{(l)}, X_{j,:}^{(l)})}_{\text{learnable}} \tag{2}$$

where $\mathfrak{C}^{(s)}$ denotes the fixed mask of the $s$-th learnable convolution, which may depend on the adjacency matrix $A$ and input edge features $E_{i,j}$. While $h^{(s)}(\cdot,\cdot) : \mathbb{R}^{f_l} \times \mathbb{R}^{f_l} \to \mathbb{R}$ can be any learnable model parametrized by $\theta^{(l,s)}$. Sometimes a learnable convolution matrix may be further row-wise normalized as $C_{i,j}^{(l,s)} \leftarrow C_{i,j}^{(l,s)} / \sum_j C_{i,j}^{(l,s)}$, for example in GAT [3]. We stick to Eq. (2) in the main paper and discuss row-wise normalization in Appendices A and E. The receptive field of a

layer of graph convolution (Eq. (1)) is defined as a set of nodes $\mathcal{R}_i^1$ whose features $\{X_{j,:}^{(l)} \mid j \in \mathcal{R}_i\}$ determines $X_{i,:}^{(l+1)}$. We re-formulate some popular GNNs into this generalized graph convolution framework; see Table 1 and Appendix A for more.

The back-propagation rule of GNNs defined by Eq. (1) is as follows,

$$\nabla_{X^{(l)}}\ell = \sum_s \left(C^{(l,s)}\right)^{\mathsf{T}} \left(\nabla_{X^{(l+1)}}\ell \odot \sigma'\left(\sigma^{-1}\left(X^{(l+1)}\right)\right)\right)\left(W^{(l,s)}\right)^{\mathsf{T}}, \tag{3}$$

which can also be understood as a form of message passing. $\sigma'$ and $\sigma^{-1}$ are the derivative and inverse of $\sigma$ respectively and $\nabla_{X^{(l+1)}}\ell \odot \sigma'\left(\sigma^{-1}(X^{(l+1)})\right)$ is the gradients back-propagated through the non-linearity.

# 3  Scalability Problem and Theoretical Framework

When a graph is large, we are forced to mini-batch the graph by sampling a subset of $b \ll n$ nodes in each iteration. Say the node indices are $i_1, \ldots, i_b$ and a mini-batch of node features is denoted by $X_B = X_{\langle i_b \rangle,:}$. To mini-batch efficiently for any model, we hope to fetch $\Theta(b)$ information to the training device, spend $\Theta(Lb)$ training time per iteration while taking $(n/b)$ iterations to traverse through the entire dataset. However, it is intrinsically difficult for most of the GNNs to meet these three scalability requirements at the same time. The receptive field of $L$ layers of graph convolution (Eq. (1)) is recursively given by $\mathcal{R}_i^L = \bigcup_{j \in \mathcal{R}_i^1} \mathcal{R}_j^{L-1}$ (starting with $\mathcal{R}_i^1 \supseteq \{i\} \cup \mathcal{N}_i$), and its size grows exponentially with $L$. Thus, to optimize on a mini-batch of $b$ nodes, we require $\Omega(bd^L)$ inputs and training time per iteration. Sampling a subset of neighbors [2, 6] for each node in each layer does not change the exponential dependence on $L$. Although layer- [7, 25] and subgraph-sampling [9, 10] may require only $\Omega(b)$ inputs and $\Omega(Lb)$ training time per iteration, they are only able to consider an exponentially small proportion of messages compared with full-graph training. Most importantly, all existing sampling methods do not support dense convolution matrices with $O(n^2)$ non-zero terms. Please see Section 5 for a detailed comparison with sampling-based scalable methods after we introduce our framework.

**Idea of dimensionality reduction.**    We aim to develop a scalable algorithm for any GNN models that can be re-formulated as Eq. (1), where the convolution matrix can be either fixed or learnable, and either sparse or dense. The major obstacle to scalability is that, for each layer of graph convolution, to compute a mini-batch of forward-passed features $X_B^{(l+1)} = X_{\langle i_b \rangle,:}^{(l+1)}$, we need $O(n)$ entries of $C_B^{(l,s)} = C_{\langle i_b \rangle,:}^{(l,s)}$ and $X^{(l)}$, which will not fit in device memory.

*Our goal is to apply a dimensionality reduction to both convolution and node feature matrices, and then apply convolution using compressed "sketches" of $C_B^{(l,s)}$ and $X^{(l)}$.* More specifically, we look for a projection matrix $R \in \mathbb{R}^{n \times k}$ with $k \ll n$, such that the product of low-dimensional sketches $\widetilde{C}_B^{(l,s)} = C_B^{(l,s)} R \in \mathbb{R}^{b \times k}$ and $\widetilde{X}^{(l)} = R^{\mathsf{T}} X^{(l)} \in \mathbb{R}^{k \times f_l}$ is approximately the same as $C_B^{(l,s)} X^{(l)}$. The approximated product (of all nodes) $\widetilde{C}^{(l,s)} \widetilde{X}^{(l)} = C^{(l,s)} R R^{\mathsf{T}} X^{(l)}$ can also be regarded as the result of using a low-rank approximation $C^{(l,s)} R R^{\mathsf{T}} \in \mathbb{R}^{n \times n}$ of the convolution matrix such that $\mathrm{rank}\left(C^{(l,s)} R R^{\mathsf{T}}\right) \leq k$. The distributional Johnson–Lindenstrauss lemma [26] (JL for short) shows the existence of such projection $R$ with $m = \Theta(\log(n))$, and the following result by Kane and Nelson [27] shows that $R$ can be chosen to quite sparse:

**Theorem 1.** *For any convolution matrix $C \in \mathbb{R}^{n \times n}$, any column vector $X_{:,a} \in \mathbb{R}^n$ of the node feature matrix $X \in \mathbb{R}^{n \times f}$ (where $a = 1, \ldots, f$) and $\varepsilon > 0$, there exists a projection matrix $R \in \mathbb{R}^{n \times k}$ (drawn from a distribution) with only an $O(\varepsilon)$-fraction of entries non-zero, such that*

$$\Pr\left(\|CRR^{\mathsf{T}}X_{:,a} - CX_{:,a}\|_2 < \varepsilon\|CX_{:,a}\|_2\right) > 1 - \delta, \tag{4}$$

*with $k = \Theta(\log(n)/\varepsilon^2)$ and $\delta = O(1/n)$.*

Now, the sketches $\widetilde{C}_B^{(l,s)}$ and $\widetilde{X}^{(l)}$ take up $O(b\log(n))$ and $\Theta(f_l \log(n))$ memory respectively and can fit into the training and inference device. The sparsity of projection matrix $R$ is favorable

because:(1) if the convolution matrix $C^{(l,s)}$ is sparse (e.g., direct-neighbor message passing where only $O(d/n)$-fraction of entries are non-zero), only an $O(\varepsilon d)$-fraction of entries are non-zero in the sketch $\widetilde{C}^{(l,s)}$; (2) During training, $\widetilde{X}^{(l)}$ is updated in a "streaming" fashion using each mini-batch's inputs $X_B^{(l)}$, and a sparse $R$ reduces the computation time by a factor of $O(\varepsilon)$. However, the projection $R$ produced following the sparse JL-lemma [27] is randomized and requires $O(\log^2(n))$ uniform random bits to sample. It is difficult to combine this with the deterministic feed-forward and back-propagation rules of neural networks, and there is no clue when and how we should update the projection matrix. Moreover, randomized projections destroy the "identity" of each node, and for learnable convolution matrices (Eq. (2)), it is impossible to compute the convolution matrix only using the sketch of features $\widetilde{X}^{(l)}$. For this idea to be useful, we need a deterministic and identity-preserving construction of the projection matrix $R \in \mathbb{R}^{n \times k}$ to avoid these added complexities.

# 4 Proposed Method: Vector Quantized GNN

**Dimensionality reduction using Vector Quantization (VQ).** A natural and widely-used method to reduce the dimensionality of data in a deterministic and identity-preserving manner is Vector Quantization [28] (VQ), a classical data compression algorithm that can be formulated as the following optimization problem:

$$\min_{R \in \{0,1\}^{n \times k}, \widetilde{X} \in \mathbb{R}^{k \times f}} \|X - R\widetilde{X}\|_F \quad \text{s.t. } R_{i,:} \in \{e_k^1, \ldots, e_k^k\}, \tag{5}$$

which is classically solved via *k-means* [28]. Here the sketch of features $\widetilde{X}$ is called the feature "codewords." $R$ is called the codeword assignment matrix, whose rows are unit vectors in $\mathbb{R}^k$, i.e., $R_{i,v} = 1$ if and only if the $i$-th node is assigned to the $v$-th cluster in *k-means*. The objective in Eq. (5) is called Within-Cluster Sum of Squares (WCSS), and we can define the relative error of VQ as $\epsilon = \|X - R\widetilde{X}\|_F / \|X\|_F$. The rows of $\widetilde{X}$ are the $k$ codewords (i.e., centroids in *k-means*), and can be computed as $\widetilde{X} = \text{diag}^{-1}(R^\mathsf{T}\mathbf{1}_n)R^\mathsf{T}X$, which is slightly different from the definition in Section 3 as a row-wise normalization of $R^\mathsf{T}$ is required. The sketch of the convolution matrix $\widetilde{C}$ can still be computed as $\widetilde{C} = CR$. In general, VQ provides us a principled framework to learn the low-dimensional sketches $\widetilde{X}$ and $\widetilde{C}$, in a deterministic and node-identity-preserving manner. However, to enable mini-batch training and inference of GNNs using VQ, three more questions need to be answered:

- How to approximate the forward-passed mini-batch features of nodes using the learned codewords?

- How to back-propagate through VQ and estimate the mini-batch gradients of nodes?

- How to update the codewords and assignment matrix along with the training of GNN?

In the following part of this section, we introduce the VQ-GNN algorithm by answering all the three questions and presenting a scalability analysis.

**Approximated forward and backward message passing.** To approximate the forward pass through a GNN layer (Eq. (1)) with a mini-batch of nodes $\langle i_b \rangle$, we can divide the messages into two categories: intra-mini-batch messages, and messages from out-of-mini-batch nodes; see the right figure of Fig. 1. Intra-mini-batch messages $C_{\text{in}}^{(l,s)} X_B^{(l)}$ can always be computed exactly, where $C_{\text{in}}^{(l,s)} = (C_B^{(l,s)})_{:,\langle i_b \rangle} \in \mathbb{R}^{b \times b}$, because they only rely on the previous layer's node features of the current mini-batch. Equipped with the codewords $\widetilde{X}^{(l)}$ and the codeword assignment of all nodes $R^{(l)}$, we can approximate the messages from out-of-mini-batch nodes as $\widetilde{C}_{\text{out}}^{(l,s)} \widetilde{X}^{(l)}$, where $\widetilde{X}^{(l)} = \text{diag}^{-1}(R^\mathsf{T}\mathbf{1}_n)R^\mathsf{T}X^{(l)}$ as defined above and $\widetilde{C}_{\text{out}}^{(l,s)} = C_{\text{out}}^{(l,s)}R$. Here, $C_{\text{out}}^{(l,s)}$ is the remaining part of the convolution matrix after removing the intra-mini-batch messages, thus $(C_{\text{out}}^{(l,s)})_{:,j} = (C_B^{(l,s)})_{:,j}\mathbb{1}\{j \in \langle i_b \rangle\}$ for any $j \in \{1, \ldots, n\}$, and $\widetilde{C}^{(l,s)}$ is the sketch of $C_{\text{out}}^{(l,s)}$. In general, we can easily approximate the forward-passed mini-batch features $X_B^{(l+1)}$ by $\widehat{X}_B^{(l+1)} = \sigma\big(\sum_s (C_{\text{in}}^{(l,s)} X_B^{(l)} + \widetilde{C}_{\text{out}}^{(l,s)} \widetilde{X}^{(l)})W^{(l,s)}\big)$.

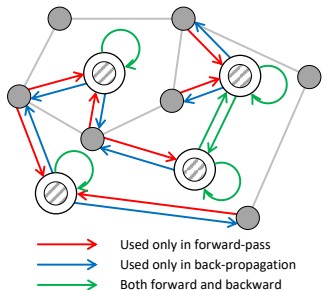

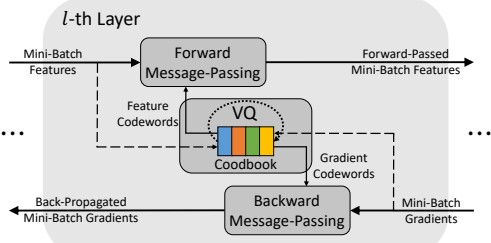

Figure 2: Three types of messages contribute to the mini-batch features and gradients. We only need "red" and "green" messages for the forward-pass. However, "blue" messages are required for back-propagation. The "red", "blue", and "green" messages are characterized by $\widetilde{C}_{\text{out}}$, $(\widetilde{C^{\mathsf{T}}})_{\text{out}}$, and $C_{\text{in}}$ respectively (Eqs. (6) and (7)).

Figure 3: For each layer, VQ-GNN estimates the forward-passed mini-batch features using the previous layer's mini-batch features and the feature codewords through approximated forward message-passing (Eq. (6)). The back-propagated mini-batch gradients are estimated in a symmetric manner with the help of gradient codewords (Eq. (7)).

However, the above construction of $\widehat{X}_B^{(l+1)}$ does not allow us to back-propagate through VQ straightforwardly using chain rules. During back-propagation, we aim at approximating the previous layer's mini-batch gradients $\nabla_{X_B^{(l)}}\ell$ given the gradients of the (approximated) output $\nabla_{\widehat{X}_B^{(l+1)}}\ell$ (Eq. (3)). Firstly, we do not know how to compute the partial derivative of $\widetilde{C}_{\text{out}}^{(l,s)}$ and $\widetilde{X}^{(l)}$ with respect to $X_B^{(l)}$, because the learning and updating of VQ codewords and assignment are *data dependent* and are usually realized by an iterative optimization algorithm. Thus, we need to go through an iterative computation graph to evaluate the partial derivative of $R^{(l)}$ with respect to $X_B^{(l)}$, which requires access to many historical features and gradients, thus violating the scalability constraints. Secondly, even if we apply some very rough approximation during back-propagation as in [29], that is, assuming that the partial derivative of $R^{(l)}$ with respect to $X_B^{(l)}$ can be ignored (i.e., the codeword assignment matrix is detached from the computation graph, known as "straight through" back-propagation), we are not able to evaluate the derivatives of codewords $\widetilde{X}^{(l)}$ because they rely on some node features out of the current mini-batch and are not in the training device. Generally speaking, designing a back-propagation rule for VQ under the mini-batch training setup is a challenging new problem.

It is helpful to re-examine what is happening when we back-propagate on the full graph. In Section 2, we see that back-propagation of a layer of convolution-based GNN can also be realized by message passing (Eq. (3)). In Fig. 2, we show the messages related to a mini-batch of nodes can be classified into three types. The "green" and "red" messages are the intra-mini-batch messages and the messages from out-of-mini-batch nodes, respectively. Apart from them, although the "blue" messages to out-of-mini-batch nodes do not contribute to the forward-passed mini-batch features, they are used during back-propagation and are an important part of the back-propagated mini-batch gradients. Since both forward-pass and back-propagation can be realized by message passing, can we approximate the back-propagated mini-batch gradients $\nabla_{X_B^{(l)}}\ell$ in a symmetric manner? We can introduce a set of gradient codewords $\widetilde{G}^{(l+1)} = \text{diag}^{-1}(R^{\mathsf{T}}\mathbf{1}_n)R^{\mathsf{T}}G^{(l+1)}$ using the same assignment matrix, where $G^{(l+1)} = \nabla_{\widehat{X}^{(l+1)}}\ell \odot \sigma'\big(\sigma^{-1}(X^{(l+1)})\big)$ is the gradients back-propagated through non-linearity. Each gradient codeword corresponds one-to-one with a feature codeword since we want to use only one assignment matrix $R$. Each pair of codewords are concatenated together during VQ updates. Following this idea, we define the approximated forward and backward message passing as follows:

$$\begin{bmatrix}\widehat{X}_B^{(l+1)} \\ \cdot\end{bmatrix} = \sigma\left(\sum_s \underbrace{\begin{bmatrix} C_{\text{in}}^{(l,s)} & \widetilde{C}_{\text{out}}^{(l,s)} \\ (\widehat{C^{(l,s)\mathsf{T}}})_{\text{out}} & \mathbf{0} \end{bmatrix}}_{\substack{\text{approx. message passing} \\ \text{weight matrix } \mathscr{C}^{(l,s)}}} \underbrace{\begin{bmatrix} X_B^{(l)} \\ \widetilde{X}^{(l)} \end{bmatrix}}_{\substack{\text{mini-batch features} \\ \text{and feat. codewords}}} W^{(l,s)}\right), \qquad (6)$$

$$\begin{bmatrix}\widehat{\nabla}_{X_B^{(l)}}\ell \\ \cdot\end{bmatrix} = \sum_s \left(\mathscr{C}^{(l,s)}\right)^{\mathsf{T}} \underbrace{\begin{bmatrix} G_B^{(l+1)} \\ \widetilde{G}^{(l+1)} \end{bmatrix}}_{\substack{\text{mini-batch gradients} \\ \text{and grad. codewords}}} \left(W^{(l,s)}\right)^{\mathsf{T}}, \qquad (7)$$

where $\mathscr{C}^{(l,s)} \in \mathbb{R}^{(b+m)\times(b+m)}$ is the approximated message passing weight matrix and is shared during the forward-pass and back-propagation process. The lower halves of the left-hand side vectors of Eqs. (6) and (7) are used in neither the forward nor the backward calculations and are never calculated during training or inference. The approximated forward and backward message passing enables the end-to-end mini-batch training and inference of GNNs and is the core of our VQ-GNN framework.

**Error-bounds on estimated features and gradients.**    We can effectively upper bound the estimation errors of mini-batch features and gradients using the relative error $\epsilon$ of VQ under some mild conditions. For ease of presentation, we assume the GNN has only one convolution matrix in the following theorems.

**Theorem 2.** *If the VQ relative error of l-th layer is $\epsilon^{(l)}$, the convolution matrix $C^{(l)}$ is either fixed or learnable with the Lipschitz constant of $h_{\theta^{(l)}}(\cdot) : \mathbb{R}^{2f_l} \to \mathbb{R}$ upper-bounded by $\mathcal{L}ip(h_{\theta^{(l)}})$, and the Lipschitz constant of the non-linearity is $\mathcal{L}ip(\sigma)$, then the estimation error of forward-passed mini-batch features satisfies,*

$$\|\widehat{X}_B^{(l+1)} - X_B^{(l+1)}\|_F \le \epsilon^{(l)} \cdot (1 + O(\mathcal{L}ip(h_{\theta^{(l)}})))\mathcal{L}ip(\sigma)\|C^{(l)}\|_F\|X^{(l)}\|_F\|W^{(l)}\|_F. \quad (8)$$

**Corollary 3.** *If the conditions in Theorem 2 hold and the non-linearity satisfies $|\sigma'(z)| \le \sigma'_{\max}$ for any $z \in \mathbb{R}$, then the estimation error of back-propagated mini-batch gradients satisfies,*

$$\|\widehat{\nabla}_{X_B^{(l)}}\ell - \nabla_{X_B^{(l)}}\ell\|_F \le \epsilon^{(l)} \cdot (1 + O(\mathcal{L}ip(h_{\theta^{(l)}}))\sigma'_{\max}\|C^{(l)}\|_F\|\nabla_{X^{(l+1)}}\ell\|_F\|W^{(l)}\|_F. \quad (9)$$

Note that the error bounds rely on the Lipschitz constant of $h(\cdot)$ when the convolution matrix is learnable. In practice, we can Lipshitz regularize GNNs like GAT [3] without affecting their performance; see Appendix E.

**VQ-GNN: the complete algorithm and analysis of scalability.**    The only remaining question is how to update the learned codewords and assignments during training? In this paper, we use the VQ update rule proposed in [29], which updates the codewords as exponential moving averages of the mSeini-batch inputs; see Appendix E for the detailed algorithm. We find such an exponential moving average technique suits us well for the mini-batch training of GNNs and resembles the *online k-means* algorithm. See Fig. 3 for the schematic diagram of VQ-GNN, and the complete pseudo-code is in Appendix E.

With VQ-GNN, we can mini-batch train and perform inference on large graphs using GNNs, just like a regular neural network (e.g., MLP). We have to maintain a small codebook of $k$ codewords and update it for each iteration, which takes an extra $O(Lkf)$ memory and $O(Lnkf)$ training time per epoch, where $L$ and $f$ are the numbers of layers and (hidden) features of the GNN respectively. We can effectively preserve all messages related to a mini-batch while randomly sampling nodes from the graph. The number of intra-mini-batch messages is $O(b^2d/n)$ when the nodes are sampled randomly. Thus we only need to pass $O(b^2d/n + bk)$ messages per iteration and $O(bd + nk)$ per epoch. In practice, when combined with techniques including product VQ and implicit whitening (see Appendix E), we can further improve the stability and performance of VQ-GNN. These theoretical and experimental analyses justify the efficiency of the proposed VQ-GNN framework.

## 5    Related Work

In this section, we review some of the recent scalable GNN methods and analyze their theoretical memory and time complexities, with a focus on scalable algorithms that can be universally applied to a variety of GNN models (like our VQ-GNN framework), including NS-SAGE[2] [2], Cluster-GCN [9], and GraphSAINT [10]. We consider GCN here as the simplest benchmark. For a GCN with $L$ layers and $f$-dimensional (hidden) features in each layer, when applied to a sparse graph with $n$ nodes and $m$ edges (i.e., average degree $d = m/n$) for "full-graph" training and inference: the memory usage is $O(Lnf + Lf^2)$ and the training/inference time is $O(Lmf + Lnf^2)$. We further assume the graph is large and consider the training and inference device memory is $O(b)$ where $b$ is the mini-batch

---

[2]We call the neighbor sampling method in [2] NS-SAGE and the GNN model in the same paper SAGE-Mean to avoid ambiguity.

Table 2: Memory and time complexities of sampling-based methods and our approach; see Section 5 for details.

| Scalable Method | Memory Usage | Pre-computation Time | Training Time | Inference Time |
|---|---|---|---|---|
| NS-SAGE | $O(br^L f + Lf^2)$ | — | $O(nr^L f + nr^{L-1} f^2)$ | |
| Cluster-GCN | $O(Lbf + Lf^2)$ | $O(m)$ | $O(Lmf + Lnf^2)$ | $O(nd^L f + nd^{L-1} f^2)$ |
| GraphSAINT-RW | $O(L^2 bf + Lf^2)$ | — | $O(L^2 nf + L^2 nf^2)$ | |
| **VQ-GNN (Ours)** | $O(Lbf + Lf^2 + Lkf)$ | — | $O(Lbdf + Lnf^2 + Lnkf)$ | $O(Lbdf + Lnf^2)$ |

size (i.e., the memory bottleneck limits the mini-batch size), and generally $d \ll b \ll n \ll m$ holds. We divide sampling baselines into three categories, and the complexities of selected methods are in Table 2. See Appendix D for more related work discussions.

**Neighbor-sampling.** Neighbor sampling scheme chooses a subset of neighbors in each layer to reduce the amount of message passing required. NS-SAGE [2] samples $r$ neighbors for each node and only aggregates the messages from the sampled node. For a GNN with $L$ layers, $O(br^L)$ nodes are sampled in a mini-batch, which leads to the complexities growing exponentially with the number of layers $L$; see Table 2. Therefore, NS-SAGE is not scalable on large graphs for a model with an arbitrary number of layers. NS-SAGE requires all the neighbors to draw non-stochastic predictions in the inference phase, resulting in a $O(d^L)$ inference time since we cannot fit $O(n)$ nodes all at once to the device. VR-GCN [6] proposes a variance reduction technique to further reduce the size $r$ of sampled neighbors. However, VR-GCN requires a $O(Lnf)$ side memory of all the nodes' hidden features and suffers from this added memory complexity.

**Layer-sampling.** These methods perform node sampling independently in each layer, which results in a constant sample size across all layers and limits the exponential expansion of neighbor size. FastGCN [7] applies importance sampling to reduce variance. Adapt [25] improves FastGCN by an additional sampling network but also incurs the significant overhead of the sampling algorithm.

**Subgraph-sampling.** Some proposed schemes sample a subgraph for each mini-batch and perform forward and backward passes on the same subgraph across all layers. Cluster-GCN [9] partitions a large graph into several densely connected subgraphs and samples a subset of subgraphs (with edges between clusters added back) to train in each mini-batch. Cluster-GCN requires $O(m)$ pre-computation time and $O(bd)$ time to recover the intra-cluster edges when loading each mini-batch. GraphSAINT [10] samples a set of nodes and takes the induced subgraph for mini-batch training. We consider the best-performing variant, GraphSAINT-RW, which uses $L$ steps of random walk to induce subgraph from $b$ randomly sampled nodes. $O(Lb)$ nodes and edges are covered in each of the $n/b$ mini-batches. Although $O(Ln)$ nodes are sampled with some repetition in an epoch, the number of edges covered (i.e., messages considered in each layer of a GNN) is also $O(Ln)$ and is usually much smaller than $m$. GraphSAINT-Node, which randomly samples nodes for each mini-batch, does not suffer from this $L$ factor in the complexities. However, its performance is worse than GraphSAINT-RW's. Like NS-SAGE and some other sampling methods, Cluster-GCN and GraphSAINT-RW cannot draw predictions on a randomly sampled subgraph in the inference phase. Thus they suffer from the same $O(d^L)$ inference time complexity as NS-SAGE; see Table 2.

## 6 Experiments

In this section, we verify the efficiency, robustness, and universality of VQ-GNN using a series of experiments. See Appendix F for implementation details and Appendix G for ablation studies and more experiments.

**Scalability and efficiency: memory usage, convergence, training and inference time.** We summarize the *memory usage* of scalable methods and our VQ-GNN framework in Table 3. Based on the implementations of the PyG library [30], memory consumption of GNN models usually grows linearly with respect to both the number of nodes and the number of edges in a mini-batch. On the *ogbn-arxiv* benchmark, we fix the number of gradient-descended nodes and the number of messages passed in a mini-batch to be $85$K and $1.5$M respectively for fair comparisons among the sampling

Table 3: Peak memory usage. Evaluated when fixing the number of gradient-descended nodes or the number of messages passed per mini-batch to be the same. Tested for GCN and SAGE-Mean on the *ogbn-arxiv* benchmark.

| Fixed GNN Model | 85K nodes per batch | | 1.5M messages passed per batch | |
|---|---|---|---|---|
| | GCN | SAGE-Mean | GCN | SAGE-Mean |
| NS-SAGE | — | 1140.3 MB | — | 953.7 MB |
| Cluster-GCN | 501.5 MB | 514.1 MB | 757.4 MB | 769.3 MB |
| GraphSAINT-RW | 526.5 MB | 519.2 MB | 661.6 MB | 650.4 MB |
| **VQ-GNN (Ours)** | 758.0 MB | 801.8 MB | 485.5 MB | 508.5 MB |

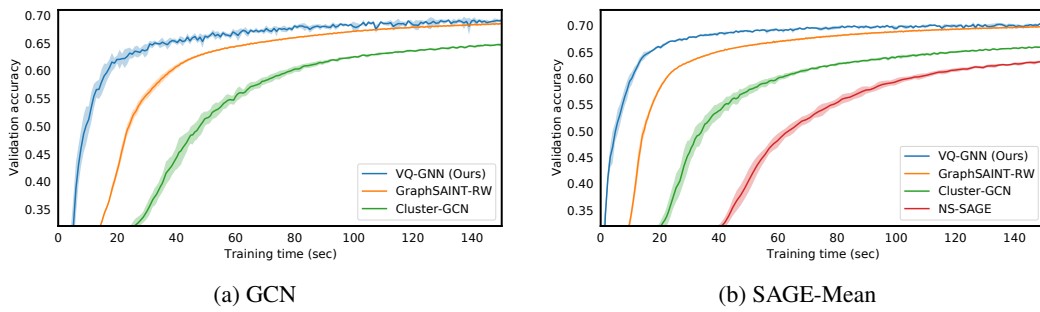

(a) GCN          (b) SAGE-Mean

Figure 4: Convergence curves (validation accuracy vs. training time). Mini-batch size and learning rate are kept the same. Tested for GCN and SAGE-Mean on the *ogbn-arxiv* benchmark.

methods and our approach. VQ-GNN might require some small extra memory when provided with the same amount of nodes per batch, which is the cost to retain all the edges from the original graph. However, our VQ-GNN framework can effectively preserve all the edges connected to a mini-batch of nodes (i.e., never drop edges); see Fig. 1. Thus when we fix the number of messages passed per batch, our method can show significant memory efficiency compared with the sampling baselines.

Fig. 4 shows the convergence comparison of various scalability methods, where we see VQ-GNN is superior in terms of the *convergence speed* with respect to the *training time*. When training GCN and SAGE-Mean on the *ogbn-arxiv* benchmark for a specific amount of time (e.g., $100$ s), the validation performance of VQ-GNN is always the highest. The training time in Fig. 4 excludes the time for data loading, pre-processing, and validation set evaluation.

Our VQ-GNN approach also leads to compelling *inference* speed-ups. Despite the training-efficiency issues of GNNs, conducting inference on large-scale graphs suffers from some unique challenges. According to our discussions in Section 5, and following the standard implementations provided by the Open Graph Benchmark (OGB) [5], the three sampling-based baselines (which share the same inference procedure) require all of the $L$-hop neighbors of the mini-batch nodes to lie on the device at once during the inference phase. The inference time of SAGE-Mean trained with sampling-methods on the *ogbn-arxiv* benchmark is $1.61$ s, while our method can accelerate inference by an order of magnitude and reduce the inference time to $0.40$ s.

**Performance comparison across various datasets, settings, and tasks.** We validate the efficacy of our method on various benchmarks in Table 4. The four representative benchmarks are selected because they have very different types of datasets, settings, and tasks. The *ogbn-arxiv* benchmark is a common citation network of arXiv papers, while *Reddit* is a very dense social network of Reddit posts, which has much more features per node and larger average node degree; see Table 6 in Appendix F for detailed statistics of datasets. *PPI* is a node classification benchmark under the inductive learning setting, i.e., neither attributes nor connections of test nodes are present during training, while the other benchmarks are all transductive. VQ-GNN can be applied under the inductive setting with only one extra step: during the inference stage, we now need to find the codeword assignments (i.e., the nearest codeword) of the test nodes before making predictions since we have no access to the test nodes during training. Neither the learned codewords nor the GNN parameters are updated during inference. *ogbl-collab* is a link prediction benchmark where the labels and loss are intrinsically different.

It is very challenging for a scalable method to perform well on all benchmarks. In Table 4, we confirm that VQ-GNN is more *robust* than the three sampling-based methods. Across the four benchmarks,

Table 4: Performance comparison between sampling-based baselines and our approach, VQ-GNN.

| Task
Benchmark | Node Classification (Transductive)
*ogbn-arxiv* (Acc.±std.) | | | Node Classification (Transductive)
*Reddit* (Acc.±std.) | | |
|---|---|---|---|---|---|---|
| GNN Model | GCN | SAGE-Mean | GAT | GCN | SAGE-Mean | GAT |
| "Full-Graph" | $.7029 \pm .0036$ | $.6982 \pm .0038$ | $.7097 \pm .0035$ | OOM[2] | OOM[2] | OOM[2] |
| NS-SAGE. | NA[1] | $.7094 \pm .0060$ | $.7123 \pm .0044$ | NA[1] | $.9615 \pm .0089$ | $.9426 \pm .0043$ |
| Cluster-GCN | $.6805 \pm .0074$ | $.6976 \pm .0049$ | $.6960 \pm .0062$ | $.9264 \pm .0034$ | $.9456 \pm .0061$ | $.9380 \pm .0055$ |
| GraphSAINT-RW | $.7079 \pm .0057$ | $.6987 \pm .0039$ | $.7117 \pm .0032$ | $.9225 \pm .0057$ | $.9581 \pm .0074$ | $.9431 \pm .0067$ |
| **VQ-GNN (Ours)** | $.7055 \pm .0033$ | $.7028 \pm .0047$ | $.7043 \pm .0034$ | $.9399 \pm .0021$ | $.9449 \pm .0024$ | $.9438 \pm .0059$ |

| Task
Benchmark | Node Classification (Inductive)
*PPI* (F1-score[3]±std.) | | | Link Prediction (Transductive)
*ogbl-collab* (Hits@50±std.) | | |
|---|---|---|---|---|---|---|
| GNN Model | GCN | SAGE-Mean | GAT | GCN | SAGE-Mean | GAT |
| "Full-Graph" | $.9173 \pm .0039$ | $.9358 \pm .0046$ | $.9722 \pm .0035$ | $.4475 \pm .0107$ | $.4810 \pm .0081$ | $.4048 \pm .0125$ |
| NS-SAGE. | NA[1] | $.9121 \pm .0033$ | $.9407 \pm .0025$ | NA[1] | $.4776 \pm .0041$ | $.3499 \pm .0142$ |
| Cluster-GCN | $.8852 \pm .0066$ | $.8810 \pm .0091$ | $.9051 \pm .0077$ | $.4068 \pm .0096$ | $.3486 \pm .0216$ | $.3905 \pm .0152$ |
| GraphSAINT-RW | $.9110 \pm .0057$ | $.9382 \pm .0074$ | $.9612 \pm .0042$ | $.4368 \pm .0169$ | $.3359 \pm .0128$ | $.3489 \pm .0114$ |
| **VQ-GNN (Ours)** | $.9549 \pm .0058$ | $.9578 \pm .0019$ | $.9737 \pm .0033$ | $.4316 \pm .0134$ | $.4673 \pm .0164$ | $.4102 \pm .0099$ |

[1] NS-SAGE sampling method is not compatible with the GCN backbone.   [2] "OOM" refers to "out of memory". The "full-graph" training on the *Reddit* benchmark requires more than 11 GB of memory.   [3] The *PPI* benchmark comes with multiple labels per node, and the evaluation metric is F1 score instead of accuracy.

VQ-GNN can always achieve performance similar with or better than the oracle "full-graph" training performance, while the other scalable algorithms may suffer from performance drop in some cases. For example, NS-SAGE fails when training GAT on *ogbl-collab*, Cluster-GCN consistently falls behind on *PPI*, and GraphSAINT-RW's performance drops on the *ogbl-collab* when using SAGE-Mean and GAT backbones. We think the robust performance of VQ-GNN is its unique value among the many other scalable solutions. VQ-GNN framework is robust because it provides bounded approximations of "full-graph" training (Theorem 2 and Corollary 3), while most of the other scalable algorithms do not enjoy such a theoretical guarantee. VQ-GNN is also *universal* to various backbone models, including but not limited to GCN, SAGE-Mean, and GAT shown here; see Appendix G for more experiments on GNNs that utilize multi-hop neighborhoods and global context, e.g., graph transformers.

## 7    Conclusion

**Summary of our framework: strengths, weaknesses, future directions, and broader impacts.** This paper introduced the proposed VQ-GNN framework, which can scale most state-of-the-art GNNs to large graphs through a principled and fundamentally different approach compared with sampling-based methods. We have shown both theoretically and experimentally that our approach is efficient in memory usage, training and inference time, and convergence speed. VQ-GNN can be universally applied to most GNN models and different graph learning tasks and can equivalently scale-up GNNs utilizing many-hops-away or global context for each layer. However, the performance of VQ-GNN relies on the quality of approximation provided by VQ. In practice, for VQ to work adequately in GNN, a set of techniques are necessary. Because of the limited time, we did not heuristically explore all possible techniques or optimize the VQ design. Given that our preliminary design of VQ in GNN already achieved competitive performance compared with the state-of-the-art sampling baselines, we hypothesize that further optimization of VQ design could improve performance. We hope our work opens up promising new avenues of research for scaling up GNNs, which also has the potential to be applied to other data domains wherever the size of a single sample is large, e.g., long time-series or videos. Considering broader impacts, we view our work mainly as a methodological and theoretical contribution, which paves the way for more resource-efficient graph representation learning. We envision our methodological innovations can enable more scalable ways to do large-network analysis for social good. However, progress in graph embedding learning might also trigger other hostile social network analyses, e.g., extracting fine-grained user interactions for social tracking.

## Acknowledgments and Disclosure of Funding

Goldstein, Kong, and Chen were supported by the Office of Naval Research, AFOSR MURI program, the DARPA Young Faculty Award, and the National Science Foundation Division of Mathematical Sciences. Additional support was provided by Capital One Bank and JP Morgan Chase. Huang and Ding were supported by a startup fund from the Department of Computer Science of the University of Maryland, National Science Foundation IIS-1850220 CRII Award 030742-00001, DOD-DARPA-Defense Advanced Research Projects Agency Guaranteeing AI Robustness against Deception (GARD), Air Force Material Command, and Adobe, Capital One and JP Morgan faculty fellowships. Li and Dickerson were supported in part by NSF CAREER Award IIS-1846237, NSF D-ISN Award #2039862, NSF Award CCF-1852352, NIH R01 Award NLM-013039-01, NIST MSE Award #20126334, DARPA GARD #HR00112020007, DoD WHS Award #HQ003420F0035, ARPA-E Award #4334192 and a Google Faculty Research Award.

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
