# Supplementary Material

## A  Generalized Graph Convolution Framework

In this section, we present more results and discussions regarding the common generalized graph convolution framework in Section 2.

**Summary of GNNs re-formulated into the common graph convolution framework.**  As stated in Section 2, most GNNs can be interpreted as performing message passing on node features, followed by feature transformation and an activation function (Eq. (6)), which is known as the common "generalized graph convolution" framework. We list more GNN models that fall into this framework in the following Table 5.

Table 5: Summary of more GNNs re-formulated as generalized graph convolution.

| Model Name | Design Idea | Conv. Matrix Type | # of Conv. | Convolution Matrix |
|---|---|---|---|---|
| GIN[1] [4] | WL-Test | Fixed + Learnable | 2 | $\begin{cases} C^{(1)} = A \\ \mathfrak{C}^{(2)} = I_n \text{ and } h_{\epsilon^{(l)}}^{(2)} = 1 + \epsilon^{(l)} \end{cases}$ |
| ChebNet[2] [12] | Spectral Conv. | Learnable | order of poly. | $\begin{cases} \mathfrak{C}^{(1)} = I_n, \mathfrak{C}^{(2)} = 2L/\lambda_{\max} - I_n, \\ \mathfrak{C}^{(s)} = 2\mathfrak{C}^{(2)}\mathfrak{C}^{(s-1)} - \mathfrak{C}^{(s-2)} \\ \text{and } h_{\theta^{(s)}}^{(s)} = \theta^{(s)} \end{cases}$ |
| GDC[3] [18] | Diffusion | Fixed | 1 | $C = S$ |
| Graph Transformers[4] [15–17] | Self-Attention | Learnable | # of heads | $\begin{cases} \mathfrak{C}_{i,j}^{(s)} = 1 \text{ and } h_{(W_Q^{(l,s)}, W_K^{(l,s)})}^{(s)}(X_{i,:}^{(l)}, X_{j,:}^{(l)}) \\ = \exp\left(\frac{1}{\sqrt{d_{k,l}}}(X_{i,:}^{(l)}W_Q^{(l,s)})(X_{j,:}^{(l)}W_K^{(l,s)})^{\mathsf{T}}\right) \end{cases}$ |

[1] The weight matrices of the two convolution supports are the same, $W^{(l,1)} = W^{(l,2)}$. [2] Where normalized Laplacian $L = I_n - D^{-1/2}AD^{-1/2}$ and $\lambda_{\max}$ is its largest eigenvalue, which can be approximated as 2 for a large graph. [3] Where $S$ is the diffusion matrix $S = \sum_{k=0}^{\infty} \theta_k \boldsymbol{T}^k$, for example, decaying weights $\theta_k = e^{-t}\frac{t^k}{k!}$ and transition matrix $\boldsymbol{T} = \widetilde{D}^{-1/2}\widetilde{A}\widetilde{D}^{-1/2}$. [4] Need row-wise normalization. Only describes the global self-attention layer, where $W_Q^{(l,s)}, W_Q^{(l,s)} \in \mathbb{R}^{f_l, d_{k,l}}$ are weight matrices which compute the queries and keys vectors. In contrast to GAT, all entries of $\mathfrak{C}_{i,j}^{(l,s)}$ are non-zero. Different design of Graph Transformers [15–17] use graph adjacency information in different ways, and is not characterized here, see the original papers for details.

**Receptive fields.**  The model in the top part of Table 5 (GIN) and the models in Table 1 in Section 2 (GCN, SAGE-Mean, and GAT) follow direct-neighbor message passing, and their single-layer receptive fields, defined as a set of nodes $\mathcal{R}_i^1$ whose features $\{X_{j,:}^{(l)} \mid j \in \mathcal{R}_i\}$ determines $X_{i,:}^{(l+1)}$, are exactly $\mathcal{R}_i^1 = \{i\} \cup \mathcal{N}_i$, where $\mathcal{N}_i$ is the set of direct neighbors of node $i$. The models in the bottom part of Table 5 (ChebNet, GDC, Transformer) can utilize many-hops-away or gloabl context each layer, and their single-layer receptive field $\mathcal{R}_i^1 \supseteq \{i\} \cup \mathcal{N}_i$.

**GNNs that cannot be defined as graph convolution.**  Some GNNs, including Gated Graph Neural Networks [31] and ARMA Spectral Convolution Networks [14] cannot be re-formulated into this common graph convolution framework because they rely on either Recurrent Neural Networks (RNNs) or some iterative processes, which are out of the paradigm of message passing.

**Further row-wise normalization of a learnable convolution matrix.**  Sometimes a learnable convolution matrix many be further row-wise normalized as $C_{i,j}^{(l,s)} \leftarrow C_{i,j}^{(l,s)} / \sum_j C_{i,j}^{(l,s)}$. This is required for self-attention-based GNNs, including GAT and Graph Transformers. This normalization process will not affect the single-layer receptive fields of the two models and can be handled by a small modification to the VQ-GNN algorithm; see Appendix E.

## B  Proofs of Theoretical Results

This section provides the formal proofs of the theoretical results in the paper.

**Proof of Theorem 1.** The proof is based on the sparse distributional Johnson–Lindenstrauss (JL) lemma in [27], which is stated as the following Lemma 1 for completeness.

**Lemma 1.** *For any integer $n > 0$, and any $0 < \varepsilon$, $\delta < 1/2$, there exists a probability distribution of $R \in \mathbb{R}^{n \times k}$ where $R$ has only an $O(\varepsilon)$-fraction of non-zero entries, such that for any $x \in \mathbb{R}^d$,*

$$\Pr\left((1 - \varepsilon)\|x\|_2 \leq \|R^\mathsf{T} x\|_2 \leq (1 + \varepsilon)\|x\|_2\right) > 1 - \delta', \tag{10}$$

*with $k = \Theta(\varepsilon^{-2} \log(1/\delta))$ and thus $\delta' = O(e^{-\varepsilon^2 k})$.*

*Proof of Theorem 1:* From the given lemma, consider the inner product $x^\mathsf{T} y$ between any two vectors $x, y \in \mathbb{R}^n$. Using the union-bound, one can derive that,

$$\Pr\left(|x^\mathsf{T} RR^\mathsf{T} y - x^\mathsf{T} y| \leq \varepsilon |x^\mathsf{T} y|\right) > 1 - 2\delta'.$$

Now, let $x$ be any row vector of $C$ and $y$ be any column vector of $X$, i.e., $x = C_{b,:}$ and $y = X_{:,a}$ for any $b \in \{1, \ldots, n\}$ and $a \in \{1, \ldots, f\}$, then we get,

$$\Pr\left(|C_{b,:}RR^\mathsf{T} X_{:,a} - C_{b,:}X_{:,a}| \leq \varepsilon |C_{b,:}X_{:,a}|\right) > 1 - 2\delta'.$$

Therefore, by the union bound again, we have,

$$\Pr\left(\|CRR^\mathsf{T} X_{:,a} - CX_{:,a}\|_2 \leq \varepsilon\|CX_{:,a}\|_2\right)$$
$$\geq 1 - \sum_{b=1}^{n} \Pr\left(|C_{b,:}RR^\mathsf{T} X_{:,a} - C_{b,:}X_{:,a}| > \varepsilon |C_{b,:}X_{:,a}|\right) > 1 - 2n\delta'.$$

Now we denote $\delta = 2n\delta'$. The required $\delta = O(1/n)$ can be achieved when $k = \Theta(\log(n)/\varepsilon^2)$. Thus we finally obtain,

$$\Pr\left(\|CRR^\mathsf{T} X_{:,a} - CX_{:,a}\|_2 < \varepsilon\|CX_{:,a}\|_2\right) > 1 - \delta,$$

with $k = \Theta(\log(n)/\varepsilon^2)$ and $\delta = O(1/n)$, which concludes the proof. $\square$

**Proof of Theorem 2.** The proof is a direct application of the VQ relative-error bound and the Lipschitz continuity properties.

*Proof of Theorem 2:* If we denote the learnable convolution matrix calculated using the approximated features (i.e., feature codewords) by $C'^{(l)}$, then we have,

$$\|\widehat{X}_B^{(l+1)} - X_B^{(l+1)}\|_F \leq \mathcal{L}ip(\sigma)\|C'^{(l)}R \operatorname{diag}^{-1}(R^\mathsf{T} \mathbf{1}_n)R^\mathsf{T} X^{(l)}W^{(l)} - C^{(l)}X^{(l)}W^{(l)}\|_F$$
$$\leq \mathcal{L}ip(\sigma)\|C'^{(l)}R \operatorname{diag}^{-1}(R^\mathsf{T} \mathbf{1}_n)R^\mathsf{T} X^{(l)} - C^{(l)}X^{(l)}\|_F\|W^{(l)}\|_F,$$

where

$$\|C'^{(l)}R \operatorname{diag}^{-1}(R^\mathsf{T} \mathbf{1}_n)R^\mathsf{T} X^{(l)} - C^{(l)}X^{(l)}\|_F$$
$$\leq \|C'^{(l)} - C^{(l)}\|_F\|R \operatorname{diag}^{-1}(R^\mathsf{T} \mathbf{1}_n)R^\mathsf{T} X^{(l)}\|_F \tag{11}$$
$$+ \|C^{(l)}\|_F\|X^{(l)} - R \operatorname{diag}^{-1}(R^\mathsf{T} \mathbf{1}_n)R^\mathsf{T} X^{(l)}\|_F.$$

The relative error $\epsilon^{(l)}$ of VQ of the $l$-th layer is defined as,

$$\|X^{(l)} - R \operatorname{diag}^{-1}(R^\mathsf{T} \mathbf{1}_n)R^\mathsf{T} X^{(l)}\|_F \leq \epsilon^{(l)}\|X^{(l)}\|_F.$$

Thus the second term on the right-hand side of Eq. (11) satisfies,

$$\|C^{(l)}\|_F\|X^{(l)} - R \operatorname{diag}^{-1}(R^\mathsf{T} \mathbf{1}_n)R^\mathsf{T} X^{(l)}\|_F \leq \epsilon^{(l)} \cdot \|C^{(l)}\|_F\|X^{(l)}\|_F$$

For the first term on the right-hand side of Eq. (11) satisfies, first note that,

$$\|R \operatorname{diag}^{-1}(R^\mathsf{T} \mathbf{1}_n)R^\mathsf{T}\|_F = \sqrt{k}$$

is a constant where $k$ is the number of codewords.

And $C'^{(l)}$ is different to $C^{(l)}$, because the convolution matrix is learnable. We have

$$C_{i,j}^{(l)} = \mathfrak{C}_{i,j} h_{\theta^{(l)}}(X_{i,:}, X_{j,:}) \quad \text{and} \quad C_{i,j}'^{(l)} = \mathfrak{C}_{i,j} h_{\theta^{(l)}}(\widetilde{X}_{i,:}^{(l)}, \widetilde{X}_{j,:}^{(l)}),$$

where $\widetilde{X}^{(l)} = \text{diag}^{-1}(R^\mathsf{T} \mathbf{1}_n) R^\mathsf{T} X^{(l)}$.

Therefore, using the Lipschitz constant of $h_{\theta^{(l)}}$, because for any $i \in \{1, \dots, n\}$,

$$\|X_{i,:} - \widetilde{X}_{i,:}^{(l)}\|_2 \le \|X^{(l)} - R\,\text{diag}^{-1}(R^\mathsf{T} \mathbf{1}_n) R^\mathsf{T} X^{(l)}\|_F \le \epsilon^{(l)} \|X^{(l)}\|_F,$$

we have for any $i, j \in \{1, \dots, n\}$,

$$|C_{i,j}'^{(l)} - C_{i,j}^{(l)}| \le 2|\mathfrak{C}_{i,j}| \cdot \mathcal{L}ip(h_{\theta^{(l)}}) \epsilon^{(l)} \|X^{(l)}\|_F$$

Summing up for all $(i, j) \in \{1, \dots, n\}^2$, we have, for the first term on the right-hand side of Eq. (11),

$$\|C'^{(l)} - C^{(l)}\|_F \le 2\|\mathfrak{C}\|_F \cdot \mathcal{L}ip(h_{\theta^{(l)}}) \epsilon^{(l)} \|X^{(l)}\|_F$$

Combining these two inequalities, we finally have,

$$\|\widehat{X}_B^{(l+1)} - X_B^{(l+1)}\|_F \le \epsilon^{(l)} \cdot (1 + O(\mathcal{L}ip(h_{\theta^{(l)}}))) \mathcal{L}ip(\sigma) \|C^{(l)}\|_F \|X^{(l)}\|_F \|W^{(l)}\|_F.$$

which concludes the proof. $\qquad\square$

**Proof of Corollary 3**    The proof is similar to the proof of Theorem 2.

*Proof of Corollary 3:*    This time, we use the message passing equation for the back-propagation process, Eq. (3), restated as follows,

$$\nabla_{X^{(l)}} \ell = \sum_s \left(C^{(l,s)}\right)^\mathsf{T} \left(\nabla_{X^{(l+1)}} \ell \odot \sigma'\left(\sigma^{-1}(X^{(l+1)})\right)\right) \left(W^{(l,s)}\right)^\mathsf{T}.$$

Note that,

$$\left\|\nabla_{X^{(l+1)}} \ell \odot \sigma'\left(\sigma^{-1}(X^{(l+1)})\right)\right\|_F \le \sigma'_{\max} \|\nabla_{X^{(l+1)}} \ell\|_F,$$

The rest of the proof simply follows the proof of Theorem 2, we similarly obtain,

$$\|\widehat{\nabla}_{X_B^{(l)}} \ell - \nabla_{X_B^{(l)}} \ell\|_F \le \epsilon^{(l)} \cdot (1 + O(\mathcal{L}ip(h_{\theta^{(l)}}))) \sigma'_{\max} \|C^{(l)}\|_F \|\nabla_{X^{(l+1)}} \ell\|_F \|W^{(l)}\|_F,$$

which concludes the proof. $\qquad\square$

# C   More Theoretical Discussions

**Upper-bound the estimation error of learnable parameters' gradients.**    Given that the gradients of each node in each layer is approximated with bounded error (Corollary 3), it is not hard to see that the gradients of learnable parameters $W^{(l)}$ and $\theta^{(l)}$ are also estimated with bounded errors. Firstly, the gradients of $W^{(l)}$, $\nabla_{W^{(l)}} \ell$, can be calculated from $\nabla_{X^{(l+1)}} \ell$ before going into the approximated back-propagation process. Secondly, the gradients of $\theta^{(l)}$, $\nabla_{\theta^{(l)}} \ell$ has bounded estimation error as long as for any $(i, j) \in \{1, \dots, n\}^2$ that $|C_{i,j}'^{(l)} - C_{i,j}^{(l)}|$ is bounded, as we have shown in Appendix B.

**Upper-bound the size of VQ codebook.**    In order to effectively upper-bound the size of the VQ codebook, i.e., the number of reference vectors in VQ and the number of clusters in *k-means*, we generally need some extra mild assumptions on the distributions of the (hidden) features and gradients. For instance, if we assume the distributions of each feature and gradient in the $l$-th layer are *sub-Gaussian*, i.e., they have strong tail decay dominated by the tails of a Gaussian. Then, we can show the relative error of VQ $\epsilon = O(k^{-1/f})$, where $f$ is the dimensionality of features and gradients. In addition to this, if we use product VQ (see Appendix E) to utilize multiple VQ process, each working on a subset of $f_{\text{prod}} \ll f$ features and gradients, then we can bound the size of codebook as $k = O(\epsilon^{-f_{\text{prod}}})$.

**Justification of choosing VQ as the dimensionality reduction method.** We choose VQ as the method of dimensionality reduction to scale up GNNs because of the following reasons:

- Categorical constraint of VQ: We choose VQ mainly because of its categorical constraint, i.e., the rows of projection matrix $R$ are unit vectors, shown in Eq. (5) of the manuscript. Intuitively, this means each node feature vector corresponds to exactly one codeword vector at a time, and thus we can replace its feature vector with the corresponding codeword. This is what we mean by "node-identity-preserving" in Sections 3 and 4.

- Compared with PCA: PCA is not suitable when we want to compress the feature table of $n$ nodes into a compact codebook when the number of nodes $n$ is much larger than the number of features per node. Moreover, PCA is shown to have the same objective function as VQ under some settings but without the categorical constraint [32]. In PCA, each node feature is represented using the set of principal components (i.e., eigenvectors of the covariance matrix). Thus, we must use the complete set of principal components to recover each node feature vector before passing it to GNNs. Moreover, it is much harder to develop an online PCA algorithm to be used along with the training of GNNs.

- Compared with Fisher's LDA: Fisher's LDA, compared with PCA, is supervised and takes class labels into consideration. However, node classification or link prediction on a large graph is a semi-supervised learning problem, and we do not have access to all the node labels during training. We do not know how to compress the test nodes' features under the transductive learning setting, and thus LDA is not a choice.

- Compared with randomized projection and locality-sensitive hashing: VQ is a better choice than randomized projection and locality-sensitive hashing because it is deterministic. Using VQ, we do not have to deal with the extra burden introduced by stochasticity.

## D   More Discussions of Related Work

In this section, we continue the discussions of related work in Section 5 and review some other scalable or efficient methods for GNNs and beyond.

**Other efficient methods for graph representation learning: MLP-based sample models and mixed-precision approaches.** There exist some other MLP-based models, e.g., SGC [33], which requires only a one-time message passing during the pre-computation stage with $O(Lmf)$ time. However, SGC, PPRGo [34] and SIGN [35] over-simplify the GNN model and limit the expressive power. Despite the fact that they are fast, their performance is not generally comparable with other GNNs. Degree-Quant [36] and SGQuant [37] applies mixed-precision techniques specifically designed for GNNs to improve efficiency. Although Degree-Quant can reduce the runtime memory usage from 4x to 8x (SGQquant achieves 4.25-31.9x reduction), they did not provide a solution to effectively mini-batch the GNN training. We note that Degree-Quant and SGQuant do not provide means to mini-batch a large graph and are still "full-graph" training.

**Outside graph learning: VQ in neural network and efficient self-attention.** The general idea of using Vector Quantization (VQ) in a neural network is initially proposed for Variational Auto-Encoders (VAEs) in VQ-VAE [29, 38], and is later generalized to other generative modeling [39], contrastive learning [40], and self-supervised learning [41, 42]. Our work learns form their success and is one of the first attempts of applying VQ to large-scale semi-supervised learning. Our work also shares similarities with the recent advances of efficient self-attention techniques to speed up Transformer models. Linformer [43] linearizes the time and memory complexities of self-attention by projecting the inputs to a compact representation and approximating the self-attention scores by a low-rank matrix. Hamburger [44] generally discussed the update rule and back-propagation problem of VQ.

## E   Algorithm Details

**The complete pseudo-code.** Here, we present the complete pseudo-code of our VQ-GNN algorithm as Algorithm 1. Three important components of VQ-GNN, namely *approximated forward message-passing*, *approximated backward message-passing*, and *VQ Update*, are highlighted in Algorithm 1 and in Fig. 3 in Section 4.

**Algorithm 1** VQ-GNN: our proposed universal framework to scale most state-of-the-art Graph Neural Networks to large graphs using Vector Quantization. For ease of presentation, we assume the GNN has only one fixed convolution matrix.

---

**Require:** Input node features $X$, ground-truth labels $Y$
**Require:** GNN's convolution matrix $C$ (fixed convolution) or $\mathfrak{C}$ and $h(\cdot, \cdot)$ (learnable convolution)

1  **procedure** VQ-GNN$_C(X, Y)$
2     **for** $l = 0, \ldots, L-1$ **do**             ▷ *Initialization*
3        Randomly initialize GNN learnable parameters $W^{(l)}$ and $\theta^{(l)}$ and codewords $\widetilde{V}^{(l)} = \widetilde{X}^{(l)} \| \widetilde{G}^{(l+1)}$ which are feature codewords $\widetilde{X}^{(l)}$ concatenated with gradient codewords $\widetilde{G}^{(l+1)}$
4        Initialize codeword assignment $R^{(l)}$ according to the initial codewords
5     **for** epoch $t = 1, \ldots, T$ **do**
6        **for** indices $\langle i_b \rangle$ sampled from $\{1, \ldots, n\}$ **do**    ▷ *Mini-batch Training*
7           Load the mini-batch features $X_B(= X_{\langle i_b \rangle, :})$, labels $Y_B$, selected rows and columns of convolution matrix $C_B$ and $(C^{\mathsf{T}})_B$, and selected rows of each codeword assignment matrix $R_B^{(l)}$ to the training device, and set $X_B^{(0)} \leftarrow X_B$
8           **for** $l = 0, \ldots, L-1$ **do**          ▷ *Forward-Pass*
9              Compute the approximate message passing weight matrix $\mathscr{C}^{(l)}$ using $C_B, C_B^{\mathsf{T}}, R_B^{(l)}$; see Eq. (6)
10             **APPROXIMATED FORWARD MESSAGE-PASSING**: estimate next layer's features $X_B^{(l+1)}$ using previous layer's mini-batch features $X_B^{(l)}$ and feature codewords $\widetilde{X}^{(l)}$; see Eq. (6)
11          Compute $\ell = \text{LOSS}(Y_B, \hat{Y}_B)$, where the predicted labels $\hat{Y}_B = \text{SOFTMAX}(X_B^{(L)})$ (node classification) or $\text{LINKPRED}(X_B^{(L)})$ (link prediction), and set $G_B^{(L)} \leftarrow \nabla_{X_B^{(L)}} \ell$ (since no non-linearity $\sigma$ in the last GNN layer; see Eq. (7))
12          **for** $l = L-1, \ldots, 0$ **do**          ▷ *Back-Propagation*
13             **APPROXIMATED BACKWARD MESSAGE-PASSING**: estimate lower layer's gradients $G_B^{(l)}$ and $\nabla_{W^{(l)}} \ell$ using higher layer's mini-batch gradients $G_B^{(l+1)}$ and gradient codewords $\widetilde{G}^{(l+1)}$; see Eq. (7)
14          **for** $l = 0, \ldots, L-1$ **do**          ▷ *VQ Update*
15             **VQ UPDATE**: Update the concatenated codewords $\widetilde{V}^{(l)} = \widetilde{X}^{(l)} \| \widetilde{G}^{(l+1)}$ and codeword assignment $R_B^{(l)}$ using this mini-batch's concatenated feature and gradient vectors $V^{(l)} = X_B^{(l)} \| G_B^{(l+1)}$ and the old concatenated codewords
16             Synchronize the codeword assignment matrix $R^{(l)}$ stored outside the training device with the updated $R_B^{(l)}$
17          Update learnable parameters $W^{(l)}$ using the estimated gradients $\nabla_{W^{(l)}} \ell$

---

**Product VQ and VQ-update rule.** As described in Section 4, we basically follow the exponential moving average (EMA) update rule for codewords as proposed in [29]. In addition, we propose two further improvements:

1. Product VQ: we divide the $2f$-dimensional features concatenated with gradients into several $f_{\text{prod}}$-dimensional blocks. And apply VQ to each of the blocks independently in parallel.
2. Implicit whitening: we whitening transform the input features and gradients before using them for VQ update, exponentially smooth the mini-batch mean and variance of inputs, and inversely transform the learned codewords using the smoothed mean and variance.

In practice, the product VQ technique is generally required for VQ-GNN to achieve competitive performance across benchmarks and using different GNN backbones. Whitening and Lipschitz regularization are tricks that are helpful in some cases (for example, Lipschitz regularization is only helpful when training GATs). The three techniques are not introduced by us and are already used by some existing work related to VQ but outside of the graph learning community. For example, product VQ is used in [45], and whitening is used in [46].

The complete pseudo-code of VQ-update is Algorithm 2. It is important to note that in the experiments, we observed that implicit whitening helps stabilize VQ and makes it more robust across different GNN models and graph datasets. However, we observed that the smoothing of mini-batch mean and variance of gradients (which is used by the approximated backward message passing) is not

compatible with some optimization algorithms which consider the cumulative history of gradients, e.g., *Adam*. This practical incompatibility is solved by using *RMSprop* instead of *Adam*

---

**Algorithm 2** VQ-Update: our proposed algorithm to update VQ codewords and assignment using exponential moving average (EMA) estimates of codewords with implicit whitening of inputs.

---

**Require:** Input mini-batch vectors $V \in \mathbb{R}^{b \times f_{\text{prod}}}$ as a part of the $2f$-dim feature and gradients
**Require:** Exponential decay rate $\gamma$ for momentum estimates of codewords
**Require:** Exponential decay rate $\beta$ for momentum estimates of mini-batch mean and variance
**Require:** Codewords $\widetilde{V} \in \mathbb{R}^{k \times f_{\text{prod}}}$ before update
**Require:** Smoothed mean $\widetilde{\text{E}}[V]$ and variance $\widetilde{\text{Var}}[V]$ before update

1  **function** VQ-UPDATE$_{(\gamma, \beta)}(V, \widetilde{V})$           ▷ Update VQ for a $f_{\text{prod}}$-dim block
2     Whitening transform $V$ to $\bar{V}$ and get the mini-batch mean $\text{E}[V]$ and variance $\text{Var}[V]$
3     $\widetilde{\text{E}}[V] \leftarrow \widetilde{\text{E}}[V] \cdot \beta + \text{E}[V] \cdot (1 - \beta)$ (EMA update of smoothed mean)
4     $\widetilde{\text{Var}}[V] \leftarrow \widetilde{\text{Var}}[V] \cdot \beta + \text{Var}[V] \cdot (1 - \beta)$ (EMA update of smoothed variance)
5     $R_B \leftarrow$ FINDNEAREST$(\bar{V}, \widetilde{V})$ (update assignment, $(R_B)_{i,v} = 1$ if $\widetilde{V}_{v,:}$ is closest to $\bar{V}_{i,:}$)
6     $\boldsymbol{\eta} \leftarrow \boldsymbol{\eta} \cdot \gamma + R_B^{\mathsf{T}} \mathbf{1}_b \cdot (1 - \gamma)$ (momentum update of cluster sizes)
7     $\boldsymbol{\Sigma} \leftarrow \boldsymbol{\Sigma} \cdot \gamma + R_B^{\mathsf{T}} \bar{V} \cdot (1 - \gamma)$ (momentum update of cluster vector sums)
8     $\widetilde{V}_{v,:} \leftarrow \frac{1}{\boldsymbol{\eta}_v} \boldsymbol{\Sigma}_{v,:}$ (update codewords)
9     Inversely whitening transform $\widetilde{V}$ to $\widetilde{V}$ using $\widetilde{\text{E}}[V]$ and $\widetilde{\text{Var}}[V]$
10     **return** $\widetilde{V}$, $R_B$

---

**Dealing with the row-wise normalized learnable convolutions.** As mentioned in Section 2 and Appendix A, some self-attention based GNNs including GAT and Graph Transformers require further row-wise normalization of the convolution matrix as $C_{i,j}^{(l,s)} \leftarrow C_{i,j}^{(l,s)} / \sum_j C_{i,j}^{(l,s)}$. However, such a procedure is not characterized by the approximated message passing design in Section 4. Some special treatment is required to normalize the message weights passed to each target node. Actually, this normalization process can be realized by the following three steps:

1. Padding an extra dimension of ones to the messages, $X^{(l)} \leftarrow X^{(l)} \| \mathbf{1}_n$
2. Message passing using the unnormalized convolution matrix
3. Normalization by dividing the last dimension, $X_{i,1:f}^{(l+1)} \leftarrow X_{i,1:f}^{(l+1)} / X_{i,f+1}^{(l+1)}$ for any $i = 1, \ldots, n$.

In this regard, we decouple the normalization of message weights with the actual message passing process. At the cost of an extra dimension of features (and thus gradients), we can VQ the message passing process again as we did in Section 4. In practice, we found this trick works well, and the experiment results on GAT and Graph Transformer in Section 6 and Appendix G are obtained using this setup.

**Regularizing the Lipschitz constants of learnable convolutions.** Since our error bounds on the approximated features and gradients (Theorem 2 and Corollary 3) rely on the Lipschitz constant of learnable convolutions, and the "decoupled row-wise normalization" trick discussed above requires some means to control the unnormalized message weights, we have to Lipschitz regularize some learnable convolution GNNs including GAT and Graph Transformers. We follow the approach described in [47] to control the Lipschitz constant of GAT and Graph Transformer without affecting their expressive power. Please see [47] for details.

**Non-linearities, dropout, normalization.** In experiments, we found our algorithm, VQ-GNN, is compatible with any non-linearities, dropout, and additional batch- or layer-normalization layers.

**Another implementation of VQ-GNN.** Following recent parallel work, GNNAutoScale [48], it is also possible to implement the VQ-GNN similarly. We can reconstruct the node features of the out-of-mini-batch neighbors for a mini-batch using the learned codewords, and then perform forward-pass message passing between the with-in-mini-batch nodes and the reconstructed nodes. This implementation is more straightforward but may suffer from larger memory overhead when the underlying graph is dense, e.g., on the *Reddit* benchmark.

# F    Implementation Details

**Hardware specs.**    Experiments are conducted on hardware with Nvidia GeForce RTX 2080 Ti (11GB GPU), Intel(R) Xeon(R) Silver 4216 CPU @ 2.10GHz, and 32GB of RAM.

**Dataset statistics.**    Detailed statistics of all datasets used in the experiments are summarized in Table 6.

Table 6: Information and statistics of the benchmark datasets.

| Dataset | *ogbn-arxiv* | *Reddit* | *PPI* | *ogbl-collab* | *Flickr* |
|---|---|---|---|---|---|
| Task | node | node | node | link | node |
| Setting | transductive | transductive | inductive | transductive | transductive |
| Label | single | single | multiple | single | single |
| Metric | accuracy | accuracy | F1-score | hits@50 | accuracy |
| # of Nodes | 169,343 | 232,965 | 56,944 | 235,868 | 89,250 |
| # of Edges | 1,166,243 | 11,606,919 | 793,632 | 1,285,465 | 449,878 |
| # of Features | 128 | 602 | 50 | 128 | 500 |
| # of Classes | 40 | 41 | 121 | (2) | 7 |
| Label Rate | 54.00% | 65.86% | 78.86% | 92.00% | 50.00% |

We note from Table 6 that:

- *PPI* is a node classification benchmark under the inductive learning setting, i.e., neither attributes nor connections of test nodes are present during training.
- *PPI* benchmark comes with multiple labels per node, and the evaluation metric is F1 score instead of accuracy.
- *Flickr* and *Reddit* have $500$ and $602$ features per node, respectively. The increased dimensionality of input node features may be challenging for VQ-GNN because VQ has to compress higher-dimensional vectors. Moreover, *Reddit*'s average node degree is $49.8$. More memory is required for mini-batches of the same size because more messages are passed in a layer of GNN.

**Hyper-parameter setups of VQ-GNN.**    To simplify the settings, we fix the hidden dimension of GNNs to 128 and layer number to 3 across the experiments. We set the size of the VQ codewords to 1024, and its size as small as 256 should also work well. We choose RMSprop (alpha=0.99) as the optimizer, and the learning rate is fixed at 3e-3. To mitigate the error induced by VQ in the high-dimensional space, we split feature vectors into small pieces. In practice, we find that when the dimension of each piece is 4, our algorithm generally works well. When the split dimension is 4 we have 32 separate branches each layer to do the VQ. These branches are independent and can be paralleled. At the end of each layer, separated feature vectors are concatenated together to restore the original hidden dimension, and the restored feature is input to the next layer. Batch norm is used for stable training. We do not use drop out for either our method or the baselines.

**Hyper-parameter setups of other scalable methods.**    For baseline models, we follow the practice of OGB. The optimizer is Adam, and the learning rate is fixed at 1e-3. For a fair comparison with respect to memory consumption, on the `ogbn-arxiv` dataset we set hyper-parameters as below: SAGE-NS with the batch size 85K, per-layer sampling sizes [20,10,5]; ClusterGCN batch size 80, number of partitions 40; GraphSAINT-RW batch size 40K, walk-length 3, number of steps 2. We use the hyper-parameter setting for experiments in Table 3 (the fixed node setting) and Figure 4. The parameter setting ensures to traverse over all the nodes in the graph in one epoch of training. For fixed message setting in 3, we alter the batch size of SAGE-NS to 35K, ClusterGCN to 60, GraphSAINT-RW to 60K. Here the batch size for ClusterGCN is small because each batch item means a cluster group of nodes.

# G    Ablation Studies and More Experiments

**Experiments on the *Flickr* benchmark**    We conduct another set of performance comparison experiments on the *Flickr* node classification benchmark in Table 7, whose information and statistics

are listed in Table 6. As in Section 6, we can draw a similar conclusion that VQ-GNN shows more robust performance than the three scalable baselines.

Table 7: Performance comparison between sampling-based baselines and our approach, VQ-GNN, on the *Flickr* benchmark.

| Task
Benchmark | Node Classification (Transductive)
*Flickr* (Acc.±std.) | | |
|---|---|---|---|
| GNN Model | GCN | SAGE-Mean | GAT |
| "Full-Graph" | $.5209 \pm .0053$ | $.5177 \pm .0041$ | $.5156 \pm .0067$ |
| NS-SAGE. | NA | $.5165 \pm .0077$ | $.5282 \pm .0052$ |
| Cluster-GCN | $.4976 \pm .0078$ | $.4996 \pm .0045$ | $.4907 \pm .0107$ |
| GraphSAINT-RW | $.5239 \pm .0071$ | $.5040 \pm .0046$ | $.5163 \pm .0062$ |
| **VQ-GNN (Ours)** | $.5315 \pm .0031$ | $.5323 \pm .0083$ | $.5288 \pm .0054$ |

**Ablation studies** We also conduct several ablation experiments on the *ogbn-arxiv* benchmark with GCN backbone. Please note that if not mentioned otherwise, the hyper-parameter setups will follow the corresponding model and are described in Appendix F.

• **Performance vs. the number of layers**: We vary the number of layers of GCN backbone from one to five, and the performance of VQ-GNN on *ogbn-arxiv* is shown in the following table. We see that a two-layer GCN is sufficient to achieve good performance on *ogbn-arxiv* while using only one layer harms the performance. Stacking more layers will not bring any performance gain.

| # of Layers | 1 | 2 | 3 | 4 | 5 |
|---|---|---|---|---|---|
| Accuracy | 0.6599 | 0.7006 | 0.7055 | 0.7080 | 0.7037 |

• **Performance vs. codebook size**: We vary the codebook size from 64 to 4096 and fix the mini-batch size at 40K. Performance is shown as follows, where we can see the performance is not sensitive to the codebook size. Setting the codebook size to 64 is enough to achieve relatively good performance on *ogbn-arxiv*. This may indicate that the node feature distribution of *ogbn-arxiv* is sparse, which is reasonable as the 128-dimensional node features are obtained by averaging the embedding of words in the titles and abstracts of arXiv papers [5].

| Codebook Size | 64 | 256 | 1024 | 4096 |
|---|---|---|---|---|
| Accuracy | 0.6950 | 0.7011 | 0.7030 | 0.7049 |

• **Performance vs. mini-batch size**: We vary the mini-batch size from 10K to 80K, and the performance is shown as follows. We see that smaller mini-batch sizes can slightly lower the overall performance. When the mini-batch size is smaller, more messages come from out-of-mini-batch nodes (see Fig. 1), and they are approximated by the VQ codewords. This increased number of approximated messages can harm the performance. However, generally speaking, the performance is not sensitive to the mini-batch size as long as it is not small.

| Mini-batch Size | 10K | 20K | 40K | 60K | 80K |
|---|---|---|---|---|---|
| Accuracy | 0.6843 | 0.6920 | 0.7011 | 0.7055 | 0.7061 |

• **Performance vs. mini-batch sampling strategy**: We compare three different sampling strategies: (1) randomly sampling nodes, (2) randomly sampling edges, and (3) sampling using random walks

as in GraphSAINT-RW. As shown in the following table, we did not observe an obvious difference in the performance.

| Mini-batch Sampling Techniques | Sampling Nodes | Sampling Edges | Sampling using Random Walks |
|---|---|---|---|
| Accuracy | 0.7020 | 0.7034 | 0.7023 |

**Performance of VQ-GNN with Graph Transformers [15].**  Our algorithm enables graph transformer architectures to compute global attention on large graphs. For each layer, we input hidden features into VQ-GNN, Graph Transformer, and Linear modules separately and sum the output features of each module together. In this way, the holistic model can not only leverage global attention but can also absorb local information. The transformer module is adapted from [15]. We show the performance in Table 8. We find that the removal of the batch norm will mitigate the problem of overfitting, so our model does not involve batch norm.

Table 8: Performance of VQ-GNN with Graph Transformer Backbone on the *ogbn-arxiv* benchmark.

| Benchmark | *ogbn-arxiv* (Acc.±std.) |
|---|---|
| Global Attention + GAT [15] | $.7108 \pm .0055$ |