# OpenReview forum: "VQ-GNN: A Universal Framework to Scale up Graph Neural Networks using Vector Quantization"
_NeurIPS.cc/2021/Conference — NeurIPS 2021 Poster_

### Official Review · Reviewer_f3BS · 2021-07-13

**Rating:** 6
**Confidence:** 3

**Summary:**

This paper proposes a technique to improve the scalability of GNNs for large graphs. In many existing works, neighbor nodes, GNN layers, or subgraphs are subsampled to reduce the burden on GPU memory. In this paper, instead of subsampling, the vector quantization (VQ) on the embedding vector of each node is introduced to reduce the memory and ensure scalability.
Technically, this paper develops a new approximation for message passing on codebook-represented graphs.
In theoretical evaluation, the paper shows that the prediction errors due to VQ can be upper-bounded by using Lipshitz regularization.
In the experimental evaluation, the generalization performance of the proposed VW method is comparable to existing scalable GNNs.

**Ethical Concerns:**

No specific concerns

**Limitations And Societal Impact:**

Many VQ-related techniques are required to make the proposed method work satisfactorily. Discussed in Sec. 7.

**Main Review:**

## Originality:
VQ + DNNs has already been studied in the famous VQ-VAE [29] paper and its followers. However, the combination of GNNs and VQ is new to me. Reducing the dimensionality of the embedding vectors is certainly a natural approach to memory compression. If there has been no previous attempt pursuing the scalability of GNNs in this direction, then I think this paper proposes a novel and valuable idea to the GNN community.

The discussion in Sec. 3 shows that if a good dimensionality reduction matrix R can be found, we can approximate well the node feature vectors. In this paper, VQ is chosen as one of the dimensionality reduction methods because it is a ''natural and widely-used method'', but are there any other ''natural and widely-used'' dimensionality reduction method choices? For example, if we don't need to use a fixed codebook, aren't PCA and FLDA also candidates?
I expect more discussions on this issue to justify the choice of VQ.

The appendix acknowledges that the quantized representation of Degree-Quant [35] improves the memory efficiency of GNNs, but [35] does not consider scalability and therefore does not conflict with the contribution of this paper.
I'm not fully sure regarding this point.
The main objective of both this paper and [35] is to reduce the memory burden of the computational device while avoiding subsampling. This paper and [35] try to achieve the same objective with different approaches: vector quantization in this paper and numerical quantization in [35]. Then these are not orthogonal, I think.
It is very possible that I misunderstand the essences, and I would appreciate clarification on this point in the rebuttal discussion.

I think that the new algorithm for learning correctly on VQ codebook-represented graph mini-batches is a great job since it is technically non-trivial and the outcome seems reasonable.

## Quality:
Except for the details of some of the formulas, I find no major problems or concerns with the technical discussion in the main manuscript.

I did not follow the details of the proofs.

## Clarity:
In the left-hand side of equations (6.7), the lower halves of the vectors are omitted.
What variables are being addressed here? Is it a variable that is not used in either the forward or backward calculations?

Table 3 is difficult to understand for me.
In particular, the comparison of ``memory usage when the number of message passing is fixed'' What is the main message here?
I read the paper several times, but in the end, I cannot figure it out. It's a concept I haven't seen in the context of many GNN papers, so unless you can explain it more clearly, I can't read any advantage of the proposed method from this table.

L302-303 ''We summarize the training and inference time........ in Table 3." should be Table 4.

## Significance:

The main weakness of this paper is that the experimental results.

The main purpose of this paper is, in my understanding, to propose a GNN technique with scalability to large graphs. However, Table 2 says that the memory usage of the proposed method is comparable to that of existing subsampling methods. This implies that there is no large graph dataset such that the proposed method is the first and the only solution for scalable GNNs.

Another issue is the generalization performance. Unlike existing subsampling methods, the proposed method is unique in that it can retain all nodes and layers (L52-53). This paper examines the advantage of this characteristic in terms of numerical performance in Table 5. Among the four scalable GNNs, the proposed method obtains the highest performance in only one out of six cases. The best performer is NS-SAGE, which takes first place in three cases, and GraphSAINT is the runner-up, best in two cases.

This means that the proposed method does not show outstanding results or unique values among several solutions that provide scalability for GNNs.

So, why should we prepare a newly proposed method instead of using existing methods that have already been evaluated and implemented (e.g., GraphSAGE)? It is difficult to answer this question clearly, therefore it is difficult for me to recommend for acceptance in the current manuscript status.

In terms of computation time and time complexity, the proposed method is clearly superior to others (Table 2, 4). Therefore, I suggest that it would be better to focus on the point of fast computation rather than the general scalability.


(+) VQ for dimensionality reduction for GNN is (probably) new
(+) Approximate message passing for VQ codebook developed
(+) Theoretical guarantee for bounded prediction error
(-) More discussions for the choice of VQ dimensionality reduction are expected.
(-) [35] is on the totally different lines of researchers, I'm not fully sure about that.
(--) Experimental results for scalability and generalization are weak.
(-) What is the main message of Table 3?

## After feedbacks
I found the author feedbacks effectively resolve many of my misunderstandings and concerns.
I raise my evaluations to 6, weak accept. Good luck!

**Time Spent Reviewing:**

8

---

> ### Author Response · Authors · 2021-08-10
> **Response to Reviewer f3BS (2/2)**
>
> ### Concern 4: Is Degree-Quant [5][\^] orthogonal to VQ-GNN?
> > "The appendix acknowledges that the quantized representation of Degree-Quant [5] improves the memory efficiency of GNNs, but [5] does not consider scalability and therefore does not conflict with the contribution of this paper. I'm not fully sure regarding this point. The main objective of both this paper and [5] is to reduce the memory burden of the computational device while avoiding subsampling. This paper and [5] try to achieve the same objective with different approaches: vector quantization in this paper and numerical quantization in [5]. Then these are not orthogonal, I think. It is very possible that I misunderstand the essences, and I would appreciate clarification on this point in the rebuttal discussion."
>
> We believe Degree-Quant [5] and SGQuant [6] are in a different line of research.
> 1. **The wording in Appendix D is misleading.** We agree it is inappropriate and misleading to say "Degree-Quant [5] and SGQuant [6] do not tackle the scalability problem of GNN", and we will update Appendix D to clarify this. We should instead say: "although Degree-Quant [5] can reduce the runtime memory usage from 4x to 8x (SGQquant [6] achieves 4.25-31.9x reduction), they did not provide a solution to effectively mini-batch the GNN training." In this sense, [5, 6] are memory-efficient versions of "full-graph" training.
> 2. **The limitations of Degree-Quant [5] and SGQuant [6]:** As you commented, Degree-Quant [5] and SGQuant [6] applied numerical quantization techniques to reduce the memory burden while avoiding subsampling. However, they do not provide means to mini-batch a large graph and are still "full-graph" training. Thus, if someone encounters a very large graph dataset, for example, the *MAG240M-LSC* benchmark in [7], which requires at least 175GB of memory just for the node attributes, it is very likely that solely using numerical quantization [5, 6], the GNN models still cannot fit in the GPU memory (usually around 10-20GB).
> 3. **Efficiency in a different sense:** That is to say, numerical quantization [5, 6] does not change the order of memory complexity with respect to the input graph size. The memory complexities of [5, 6] are still linear in the number of nodes $n$ but with some reduced coefficients. However, things are different with our VQ-GNN framework (and the other sampling-based scalable methods). As you can see in Table 2 of the manuscript, the memory complexities of VQ-GNN and sampling-based algorithms are linear in the mini-batch size $b$ and are independent of the number of nodes $n$. Using VQ-GNN, we are free to choose a small enough batch size for a memory-constrained device. Moreover, it is possible to apply numerical quantization on top of our VQ-GNN algorithm. Rigorously speaking, we should not say Degree-Quant [5] and SGQuant [6] are completely orthogonal to our work, but [5, 6] are definitely in a different line of research.
>
> We hope these discussions clear up your confusion, and we will add them to Appendix D.
>
> ### Concern 5: Variables omitted in Eq. (6) and (7).
> > "On the left-hand side of equations (6, 7), the lower halves of the vectors are omitted. What variables are being addressed here? Is it a variable that is not used in either the forward or backward calculations?"
>
> 1. **Those variables omitted are not used in any calculations and have no meaning:** Yes, the lower halves of the left-hand side vectors of Eq. (6) and (7) are used in neither the forward nor the backward calculations. They do not have any meaning and are never calculated at any time during training or inference.
> 2. **The practical explanation:** In the practical implementation, those omitted variables serve as placeholders, and their values are refreshed by the latest codewords. For example, following Eq. (6), when we forward pass through the $l$-th layer, the lower half of the left-hand side vector of Eq. (6) is not calculated. But right before we forward pass through the ($l+1$)-th layer (just replacing $l$ by ($l+1$) in Eq. (6)), we assign the latest codewords $\widetilde{X^{(l+1)}}$ to the lower-half placeholders (now on the right-hand side of Eq. (6)). That is why we choose to write Eq. (6) and (7) in this matrix form but omit the lower halves of the left-hand side vectors.
>
> We hope you understand the formulation better now. We will update the explanations in Section 4 accordingly to make this point crystal clear.
>
> ### Concern 6: Typos.
> > "L302-303 "We summarize the training and inference time........ in Table 3." should be Table 4."
>
> Thank you for pointing out these typos, and we have fixed them.
>
> ### Remarks in this response:
> * [\*] Please note that sometimes the uncertainties of performance metrics are large, and it is hard to tell which algorithm wins by comparing the average metric. For example, when training GCN on *ogbn-arxiv* (see Table 5 of the manuscript), it is hard to say GraphSAINT-RW (0.7079±0.0057) outperforms VQ-GNN (0.7055±0.0033) because the difference is smaller than the uncertainties.
> * [\^] Please note that the references are re-numbered.
>
> ### References in this response:
>
> * [1] Chiang, W. L., Liu, X., Si, S., Li, Y., Bengio, S., & Hsieh, C. J. (2019, July). Cluster-GCN: An efficient algorithm for training deep and large graph convolutional networks. In Proceedings of the 25th ACM SIGKDD International Conference on Knowledge Discovery & Data Mining (pp. 257-266).
> * [2] Zeng, H., Zhou, H., Srivastava, A., Kannan, R., & Prasanna, V. (2019). Graphsaint: Graph sampling-based inductive learning method. arXiv preprint arXiv:1907.04931.
> * [3] Fey, M., & Lenssen, J. E. (2019). Fast graph representation learning with PyTorch Geometric. arXiv preprint arXiv:1903.02428.
> * [4] Ding, C., & He, X. (2004, July). K-means clustering via principal component analysis. In Proceedings of the twenty-first international conference on Machine learning (p. 29).
> * [5] Tailor, S. A., Fernandez-Marques, J., & Lane, N. D. (2020). Degree-Quant: Quantization-aware training for graph neural networks. arXiv preprint arXiv:2008.05000.
> * [6] Feng, B., Wang, Y., Li, X., Yang, S., Peng, X., & Ding, Y. (2020, November). SGQuant: Squeezing the Last Bit on Graph Neural Networks with Specialized Quantization. In 2020 IEEE 32nd International Conference on Tools with Artificial Intelligence (ICTAI) (pp. 1044-1052). IEEE.
> * [7] Hu, W., Fey, M., Ren, H., Nakata, M., Dong, Y., & Leskovec, J. (2021). OGB-LSC: A large-scale challenge for machine learning on graphs. arXiv preprint arXiv:2103.09430.

---

> ### Author Response · Authors · 2021-08-10
> **Response to Reviewer f3BS (1/2)**
>
> Thank you so much for your constructive comments and your time reviewing our work. We really appreciate that you recognize the merits of our work and think it could be a novel and valuable idea to the GNN community. We believe some of your concerns are due to misunderstandings. And we hope our responses could mitigate your concerns and help you re-evaluate our work. We address your questions and comments as follows.
>
> ### Concern 1: VQ-GNN does not show outstanding results or unique values.
> > "This paper examines the advantage of this characteristic in terms of numerical performance in Table 5. Among the four scalable GNNs, the proposed method obtains the highest performance in only one out of six cases. The best performer is NS-SAGE, which takes first place in three cases, and GraphSAINT is the runner-up, best in two cases. This means that the proposed method does not show outstanding results or unique values among several solutions that provide scalability for GNNs."
>
> We do not agree with you on this point. Here are our explanations:
> 1. **VQ-GNN can show outstanding results.** We agree that although VQ-GNN always achieves competitive performance, in Table 5, VQ-GNN obtains the highest performance[\*] in only one out of the six cases. We then evaluate VQ-GNN on four more graph learning benchmarks and find that VQ-GNN can outperform the baselines by a margin on the newly added *FLICKR* dataset. Please check Sections 1.4 and 3.1 of **Common Response 1** for the details. We think more experimental results are needed to draw a conclusion on the performance of VQ-GNN. **And the new experiments demonstrate that VQ-GNN can outperform the other scalability methods by a margin on some real-world datasets.**
> 2. **VQ-GNN has unique values among other scalability solutions.** We think it is not appropriate to say that VQ-GNN has no unique values in its performance sorely because it only wins[\*] one out of the six cases in Table 5. Also from Table 5 of the manuscript, we can identify another fact: VQ-GNN is more robust than the other three scalable baselines. We can see NS-SAGE's performance is much lower than VQ-GNN's on *ogbl-collab* when using the GAT backbone. Meanwhile, Cluster-GCN falls behind VQ-GNN and the other baselines by a margin on *ogbn-arxiv* when the backbone is GCN, and on *ogbl-collab* when using GCN and SAGE-Mean. Moreover, GraphSAINT-RW's performance drops by a margin on the *ogbl-collab* dataset when using SAGE-Mean and GAT backbones. However, on these two datasets in Table 5, VQ-GNN can always achieve performance similar or better than the three scalable baselines (while all of the three baselines can fall behind VQ-GNN by a margin in some cases). Now, if you take a careful look at the experimental results on the four new benchmarks (see Section 3 of **Common Response 1**), you can draw a similar conclusion that VQ-GNN shows more robust performance than the three scalable baselines. **We think this robust performance of VQ-GNN is exactly its unique value among the other scalability solutions. VQ-GNN framework is robust because it provides bounded approximations of "full-graph" training (Theorem 2 and Corollary 3), but most of the other scalable algorithms do not enjoy such a theoretical guarantee.** In this sense, it is not surprising to see VQ-GNN can usually achieve similar or even better performance than "full-graph" training, while the other scalable algorithm may fail in some cases.
>
> We hope the above arguments can mitigate your main concern.
>
> ### Concern 2: How to understand the memory advantage of VQ-GNN in Table 3.
> > "Table 3 is difficult to understand for me. In particular, the comparison of "memory usage when the number of message passing is fixed." What is the main message here? I read the paper several times, but in the end, I cannot figure it out. It's a concept I haven't seen in the context of many GNN papers, so unless you can explain it more clearly, I can't read any advantage of the proposed method from this table."
>
> We understand that this type of comparison is rarely seen in most GNN papers, and we are sorry for the confusion caused. More detailed explanations are as follows:
> 1. **Explanation of "the number of messages passed":** Here, the number of messages passed can also be understood as the effective number of edges considered. Many sampling-based techniques like Cluster-GCN [1] and GraphSAINT-RW [2] ignore some edges when mini-batching the large graph. For example, when using Cluster-GCN [1], the cut edges between the clusters are dropped during mini-batch sampling before training. In this sense, they only consider a subset of the edges in the input graph, which can harm the performance of GNNs. However, **our VQ-GNN framework can effectively preserve all the edges connected to a mini-batch of nodes (i.e., never drop edges);** see Figure 1 of the manuscript.
> 2. **Why do we compare while keeping "the number of messages passed" fixed?** For some implementations of GNNs, e.g., the PyG library [3] this paper use, the memory usage of GAT is proportional to the number of messages passed in each layer, instead of the number of nodes in the mini-batch. Hence, when we compare the memory consumption of GAT with different scalable methods, it is a bit unfair if we set the mini-batch size to be the same since the sampling-based techniques only need to consider a subset of messages at the cost of harming the performance. Hence, we also compare the memory usage of GATs when the effective number of messages passed (i.e., edges considered) is kept to be the same. From the rightmost column of Table 3, we can see that our GATs' memory consumption is more advantageous under such a setting.
>
> We hope the explanations above can answer your question, and we will add them to the manuscript to make things clearer.
>
> ### Concern 3: Justify the choice of VQ as the dimensionality reduction method.
> > "In this paper, VQ is chosen as one of the dimensionality reduction methods because it is a "natural and widely-used method", but are there any other "natural and widely-used" dimensionality reduction method choices? For example, if we don't need to use a fixed codebook, aren't PCA and FLDA also candidates? I expect more discussions on this issue to justify the choice of VQ."
>
> We choose VQ as the method of dimensionality reduction to scale up GNNs because of the following reasons:
> 1. **Categorical constraint of VQ:** We choose VQ mainly because of its categorical constraint, i.e., the rows of projection matrix $R$ are unit vectors, shown in Eq. (5) of the manuscript. Intuitively, this means each node feature vector corresponds to exactly one codeword vector at a time, and thus we can replace its feature vector with the corresponding codeword. This is what we mean by "node-identity-preserving" in L152, L156, and L167 of the manuscript.
> 2. **Compared with PCA:** PCA is not suitable when we want to compress the feature table of $n$ nodes into a compact codebook when the number of nodes $n$ is much larger than the number of features per node. Moreover, PCA is shown to have the same objective function as VQ under some settings but without the categorical constraint [4]. In PCA, each node feature is represented using the set of principal components (i.e., eigenvectors of the covariance matrix). Thus, we must use the complete set of principal components to recover each node feature vector before passing it to GNNs. Moreover, it is much harder to develop an online PCA algorithm to be used along with the training of GNNs.
> 3. **Compared with Fisher's LDA:** Fisher's LDA, compared with PCA, is supervised and takes class labels into consideration. However, node classification or link prediction on a large graph is a semi-supervised learning problem, and we do not have access to all the node labels during training. We do not know how to compress the test nodes' features under the transductive learning setting, and thus LDA is not a choice.
> 4. **Compared with randomized projection and locality-sensitive hashing:** VQ is a better choice than randomized projection and locality-sensitive hashing because it is deterministic. Using VQ, we do not have to deal with the extra burden introduced by stochasticity.
>
> Thank you for this insightful question, and we will add the above discussions to the manuscript.

---

### Official Review · Reviewer_ai2z · 2021-07-16

**Rating:** 6
**Confidence:** 3

**Summary:**


The paper presents a vector quantization method to reduce the memory consumption of the convolutional operations for scalable GNNs. The method avoids subsampling nodes in the mini batches to prevent all the messages could be kept in the message passing operations. It quantizes the convolutional matrices leveraging the dimensionality reduction method. The approximated message passing with help of the quantization codewords is shown to have low relative error theoretically under certain conditions and have comparable accuracy with the original algorithm empirically.


**Limitations And Societal Impact:**

Yes.

**Main Review:**

Positive points:
1. The motivation and challenges are well-presented and related work seems to be adequate.
2. I like the idea of applying dimensionality reduction technique to quantize the feature and convolutional matrices in the GNNs. The experimental results seem convincing, though limited experiments are provided.

Comments:
1.	The notations and explanations are not very clear, especially in the ``approximated forward and backward message passing`` part of section 4.
2.	I would prefer to see more experiments and discussion on the hyperparameters selection and their affects to the algorithm such as the selection of the reduced dimension the mini-batch sizes, and the effect of different mini-batch sampling strategies.

During the rebuttal phase, the authors add additional experiments following the suggestions and I'm satisfied with the feedback and additional clarification given by the author. Considering the theoretical/experimental impact of the paper, I'd keep my rating towards marginal acceptance.

**Time Spent Reviewing:**

3 hours

---

> ### Author Response · Authors · 2021-08-10
> **Response to Reviewer ai2z**
>
> Thank you for the thoughtful comments. We really appreciate that you like our idea of applying the dimensionality reduction technique to quantize the features and convolutional matrices in GNNs. We address your questions and comments as follows.
>
> ### Concern 1: Limited experiments are provided.
> > "The experimental results seem convincing, though limited experiments are provided."
>
> We agree that the main weakness of the manuscript is the limited experimental results. In this regard, we conducted experiments on four more graph learning benchmarks (thus, we have six datasets in total now). Please check Sections 1.3 and 3 of **Common Response 1** for the details.
>
> ### Concern 2: Notations and explanations are not very clear in Section 4.
> > "The notations and explanations are not very clear, especially in the *approximated forward and backward message-passing* part of Section 4."
>
> We are sorry for the confusion caused and will try our best to polish the notations and explanations in the 2nd half of Section 4. Generally, we use tilde $\widetilde{ }$ to indicate a variable is directly estimated using the VQ codebook, and hat $\widehat{ }$ to indicate a variable is approximated by the approximated forward or backward message passing. We will spend more space to introduce the notation conventions in Section 4. Thanks for the valuable suggestion, and we are going to update the manuscript soon.
>
> ### Concern 3: More experiments on hyper-parameter selection and their effects.
> > "I would prefer to see more experiments and discussion on the hyperparameters selection and their effects to the algorithm such as the selection of the reduced dimension the mini-batch sizes, and the effect of different mini-batch sampling strategies."
>
> Thank you for the suggestion. We have followed your suggestions and conducted new experiments to understand how the choices of reduced dimensions (i.e., codebook sizes), mini-batch sizes, and mini-batch sampling strategies affect the VQ-GNN algorithm. Please check Sections 1.2, 1.3, 2.2, 2.3, and 2.4 of **Common Response 2** for the details.

---

> > ### Comment · Reviewer_ai2z · 2021-08-31
> > **Reply to Response**
> >
> > Thanks very much for the feedback and additional experimental given by the authors. I would modify my reviews and keep my ratings. Thank you again!

---

### Official Review · Reviewer_UGM2 · 2021-07-19

**Rating:** 7
**Confidence:** 4

**Summary:**

This paper presents VQ-GNN: a scalable framework for training graph neural networks that approximates full-graph message passing with a memory of global notes that is updated during training. The authors derive approximate forward and backward passes and show that the network can preserve full-batch results while reducing computational and memory costs.

**Limitations And Societal Impact:**

The authors have addressed the limitations around the tricks used to make the VQ work well. I think it is reasonable to explore these design choices in greater detail in future work. The authors briefly discuss societal risks relating to the use of large-scale GNNs in exploitative analysis of social networks.

**Main Review:**

I think the general idea of approximating the general form of convolutional matrices with a low-rank decomposition is good. Is there a connection to the Reformer architecture (Kitaev, Kaiser et al 2020) and locality-sensity-hashing approximations to full attention matrices in Transformers?

The proposed method demonstrates substantial gains on link-prediction tasks relative to alternative scalable GNN training methods. The performance (accuracy / hits) is similar - and in some cases better - than the full-graph algorithm. Combined with the memory and training / inference speed gains, it makes for a compelling set of results.

I like the analysis of the fundamental scalability limitations of other ‘scalable’ GNN training algorithms. The highlight is the futility of managing the exponential receptive field dependence on the number of layers by dropping nodes or layers. In general the paper contains extensive theoretical results, and analyses of the method’s complexity relative to alternatives.

It would be useful to see some experimental analysis on codebook / minibatch size? For instance a plot of model performance / computational costs as a function of the minibatch or codebook size. Otherwise it’s not clear how sensitive the framework is to these factors.

It seems like a potentially important bit of experimental detail is added in the appendix [L641-646]:

“To mitigate the error induced by VQ in the high-dimensional space, we split feature vectors into small pieces. In practice, we find that when the dimension of each piece is 4, our algorithm generally works well. When the split dimension is 4 we have 32 separate branches each layer to do the VQ. These branches are independent and can be paralleled. At the end of each layer, separated feature vectors are concatenated together to restore the original hidden dimension, and the restored feature is input to the next layer

I found it a bit hard to understand this. Are the 4-dimensional vector pieces independently assigned to codebooks? If this is an important component then it would be useful to report some analysis in the paper. And the same for the other VQ-details: whitening and Lipshitz regularization. If the model is highly sensitive to these elements then that is a limitation.

Overall, the proposed VQ-GNN framework is well motivated, well-executed and has promising results. Some additional analysis would be welcome, and potentially a restructuring so that some of the important practical details (product VQ etc) are made more apparent in the main body.


**Time Spent Reviewing:**

3

---

> ### Author Response · Authors · 2021-08-10
> **Response to Reviewer UGM2**
>
> Thank you for your thoughtful and constructive comments. We really appreciate that you recognize the merits of our work. We address your questions and comments as follows.
>
> ### Concern 1: Is there a connection to Reformer [1] and locality-sensitive hashing in Transformers?
> > "Is there a connection to the Reformer architecture (Kitaev, Kaiser, et al. 2020) and locality-sensitive-hashing approximations to full attention matrices in Transformers?"
>
> No, we do not think that this work has a connection to the Reformer [1] and the idea of locality-sensitive hashing (LSH) in Transformers.
> * **LSH is intrinsically different from VQ.** Although both vector quantization and locality-sensitive hashing provide ways to reduce the dimensionality of high-dimensional data, they are intrinsically different algorithms.
> * **Compare with LSH：** We think VQ is a better choice than locality-sensitive hashing mainly because it is deterministic, and we do not have to deal with the extra burden introduced by the stochasticity. We also compare VQ with some other means of dimensionality reduction, including PCA and LDA in Concern 3 of the **Response to Reviewer f3BS**, and please take a look if interested.
> * **Related work in the Transformer context:** To sum up, we only found two papers in the Transformer context that share some similarities with this work, namely Linformer [2] and Hamburger [3]. They are discussed in Appendix D.
>
> ### Concern 2: Experimental analysis on codebook and mini-batch sizes.
> > "It would be useful to see some experimental analysis on codebook/minibatch size? For instance, a plot of model performance/computational costs as a function of the minibatch or codebook size. Otherwise, it is not clear how sensitive the framework is to these factors."
>
> Thanks for the valuable suggestion. We have followed your suggestions and conduct a set of new experiments to understand how the performance and computational costs of VQ-GNN depend on the codebook and mini-batch sizes. Please check Sections 1.2, 2.2, and 2.3 of **Common Response 2** for the new experiments on performance vs. codebook and mini-batch sizes. As for the computational cost, we think it is most important to see how memory usage is affected by these two factors since memory is the critical bottleneck to scale up GNNs. When training with GCN backbone on the *ogbn-arxiv* dataset, we vary the codebook size from 64 to 1024 and find that the peak memory usage grows from 178MB to 245MB. In practice, for the six benchmarks we use, setting the codebook size to 1024 is always sufficient, and thus the memory overhead of VQ is relatively small. If we increase the mini-batch size, the peak memory usage grows approximately linearly: 184MB when the mini-batch size is 10k, 338MB when it is 20k, and 549MB when it is 40k. In this sense, we can choose a suitable mini-batch size to fit the model into a memory-constrained device. We are sorry that we cannot post new plots here. But we will add the experimental results and discussions to the manuscript.
>
> ### Concern 3: Experimental details of VQ.
> > "I found it a bit hard to understand this. Are the 4-dimensional vector pieces independently assigned to codebooks? If this is an important component, then it would be useful to report some analysis in the paper. And the same for the other VQ details: whitening and Lipshitz regularization. If the model is highly sensitive to these elements, then that is a limitation."
>
> Yes, the 4-dimensional vector pieces are independently assigned to the codebooks. We apologize that our explanations of the practical details of VQ are not very clear in the manuscript. In practice, the product VQ technique is generally required for VQ-GNN to achieve competitive performance across benchmarks and using different GNN backbones. Whitening and Lipschitz regularization are tricks that are helpful in some cases (for example, Lipschitz regularization is only helpful when training GATs). The three techniques are not introduced by us and are already used by some existing work related to VQ but outside of the graph learning community. For example, product VQ is used in [4], and whitening is used in [5]. We agree it is important to have more discussions regarding these techniques in the paper, and we are going to update the manuscript accordingly.
>
> ### Concern 4: Make the practical details more apparent in the main body.
> > "Some additional analysis would be welcome, and potentially a restructuring so that some of the important practical details (product VQ, etc.) are made more apparent in the main body."
>
> Thanks for this valuable suggestion. We agree that it would be better if we can move more important practical details to the main body. We will carefully restructure the parts you mentioned and update the manuscript soon.
>
> ### References in this response:
> * [1] Kitaev, N., Kaiser, Ł., & Levskaya, A. (2020). Reformer: The efficient transformer. arXiv preprint arXiv:2001.04451.
> * [2] Wang, S., Li, B. Z., Khabsa, M., Fang, H., & Ma, H. (2020). Linformer: Self-attention with linear complexity. arXiv preprint arXiv:2006.04768.
> * [3] Geng, Z., Guo, M. H., Chen, H., Li, X., Wei, K., & Lin, Z. (2020, September). Is Attention Better Than Matrix Decomposition?. In International Conference on Learning Representations.
> * [4] Wu, H., & Flierl, M. (2019, November). Learning product codebooks using vector-quantized autoencoders for image retrieval. In 2019 IEEE Global Conference on Signal and Information Processing (GlobalSIP) (pp. 1-5). IEEE.
> * [5] Berthelot, D., Raffel, C., Roy, A., & Goodfellow, I. (2018). Understanding and improving interpolation in autoencoders via an adversarial regularizer. arXiv preprint arXiv:1807.07543.

---

> > ### Comment · Reviewer_UGM2 · 2021-08-31
> > **Reply to Response**
> >
> > Thank you for the detailed response. I appreciate the additional analysis on codebook sizes, and the decision to update the paper with some experimental details.

---

### Official Review · Reviewer_UTjD · 2021-07-28

**Rating:** 5
**Confidence:** 4

**Summary:**

This paper is about using vector quantization to scale up GNN. The main idea is to use quantized representations combined with a low-rank projection of graph convolution matrix to avoid the "neighbor explosion" problem of GNNs. The proposed VQ-GNN is applied for node classification and link prediction, and shows comparable results.

**Main Review:**

1) Although this paper is about scaling up the GNN training, the experiment is only on small-scale datasets, only one dataset for node classification and one for link prediction. It would be interesting to see larger scale experiments with nodes size in millions.

2) Seems this paper is about scaling up the training, it would be interesting to check the real training time vs accuracy. Why in table 4, per epoch time is faster than Cluster-GCN and GraphSAINT?   It seems to me that per epoch the proposed method will be slower as it needs additional time to update the codebook.

3) How is the performance w.r.t. number of layers? how about inductive and transductive settings?



**Time Spent Reviewing:**

3 hours

---

> ### Author Response · Authors · 2021-08-10
> **Response to Reviewer UTjD**
>
> Thank you for the constructive feedback. We address your questions and comments as follows.
>
> ### Concern 1: Large-scale benchmarks with nodes size in millions.
> > "Although this paper is about scaling up the GNN training, the experiment is only on small-scale datasets, only one dataset for node classification and one for link prediction. It would be interesting to see larger scale experiments with nodes size in millions."
>
> Thanks for the suggestion. We agree that evaluating our proposed VQ-GNN on a large-scale dataset with millions of nodes is a good idea. We have conducted some extra experiments on the *ogbl-citation2* dataset with over 2.9 million nodes. Please check Sections 1.1, 2.1, and 3 of **Common Response 1** for the details.
>
> ### Concern 2: Experimental analysis on training time vs. performance.
> > "It seems this paper is about scaling up the training. It would be interesting to check the real training time vs. accuracy."
>
> The training time of VQ-GNN is mostly affected by the mini-batch size, and the dependence is similar to any other neural network: if we increase the mini-batch size, the training time per epoch decreases. When training with GCN backbone on the *ogbn-arxiv* dataset, the training time per epoch is 0.83s when the mini-batch size is 10k, 0.73s when it is 20k, and 0.70s when it is 40k. Moreover, from the new experiments (see Section 2.3 of **Common Response 2**), we know the performance of VQ-GNN is not much affected by the mini-batch size when it is large. In this regard, it is recommended to set a large mini-batch size as long as the model can be fitted into the training device.
>
> ### Concern 3: Regarding time comparison in Table 4.
> > "Why in table 4, per epoch time is faster than Cluster-GCN and GraphSAINT? It seems to me that per epoch, the proposed method will be slower as it needs additional time to update the codebook."
>
> Indeed our proposed VQ-GNN needs some extra time to update the codebook. However, Cluster-GCN and GraphSAINT-RW also introduce some time overhead to sample the mini-batches. For Cluster-GCN, an expensive $O(m)$ overhead (where $m$ is the number of edges, see Section 5 of the manuscript) is required to partition the graph into clusters for each epoch. GraphSAINT-RW also requires a relative expensive random walk simulation process on the large graph to sample the mini-batches, which is $O(Lbd)$ where $L$ is the number of layers of GNNs and also the number of steps of random walks, $b$ is the batch size, and $d$ is the average degree. We are sorry this $O(Lbd)$ pre-computation time is not added to Table 2 of the manuscript and we have fixed this problem. We will also add the above discussions to Section 5 and the caption of Table 4.
>
> ### Concern 4: Experimental analysis on performance vs. the number of layers.
> > "How is the performance w.r.t. the number of layers?"
>
> We followed your suggestion, and the expected discussion is presented in Sections 1.1 and 2.1 of **Common Response 2**.
>
> ### Concern 5: Experiments under inductive and transductive settings.
> > "How about inductive and transductive settings?"
>
> The experiments in the manuscript (on *ogbn-arxiv* and *ogbl-collab* benchmarks) are conducted under the transductive learning setting. However, in addition to that, we have evaluated our proposed VQ-GNN on *PPI* dataset under the inductive setting. VQ-GNN can be applied to the inductive setting with only one extra step. Please see Sections 1.2, 2.2, and 3.2 of **Common Response 1** for details.

---

> > ### Comment · Reviewer_UTjD · 2021-08-31
> > **wrong understanding of comparing methods**
> >
> > First of all, thanks authors to answer all my questions and conduct extra experiments especially larger scale experiment and inductive setting as well.
> >
> > However, some answers are not correct, in particular, "Why in table 4, per epoch time is faster than Cluster-GCN and GraphSAINT? It seems to me that per epoch, the proposed method will be slower as it needs additional time to update the codebook."
> >
> > The answer for the rebuttal: For Cluster-GCN, an expensive O(m)  overhead (where m is the number of edges, see Section 5 of the manuscript) is required to partition the graph into clusters for each epoch.
> >
> > The above answer is NOT correct. For cluster GCN ( checking their algorithms), in fact it first partitions the graph and saves the partitions (just node indices so no memory overhead). For every epoch of training, it does not need to  redo clustering, and all it does is to pick the clusters--so there is no overhead for clustering at all. Similar for GraphSaint, the batching time is not related to L(number of layers). This rises my concerns/correctness of these experiment in the paper.
> >
> > Therefore I would reduce my score to 5.

---

> > > ### Author Response · Authors · 2021-09-01
> > > **We did not wrongly understand the baseline methods**
> > >
> > > We thank Reviewer UTjD for reading our rebuttal and pointing out new questions. However, we want to address that **we did not wrongly understand the baseline methods as Reviewer UTjD argued**.
> > >
> > > ## For Cluster-GCN
> > >
> > > ### Clarification
> > > It is **a writing error** to say that "we are required to partition the graph for each epoch" in the rebuttal to Reviewer UTjD, and we are really sorry about that. However, we sincerely highlight that **in the literature review of our manuscript, we have correctly summarized the time complexity of Cluster-GCN**, quoted from Line 278-279:
> > >
> > > > Cluster-GCN requires **$O(m)$ pre-computation time** and **$O(bd)$ time to recover the intra-cluster edges** when loading each mini-batch.
> > >
> > > We have clearly stated that the $O(m)$ time overhead is for pre-computation and only happens once. In Table 2 of the manuscript, this $O(m)$ pre-computational time is also listed individually, not as a part of the training time. **These serve as clear proof that we did not misunderstand Cluster-GCN in our manuscript and experiments**.
> > >
> > > ### Explanation to the Concern 3
> > > Actually, in our response to Concern 3, what we would like to address is the $O(bd)$ time overhead of recovering the intra-cluster edges for each mini-batch (where $b$ is the mini-batch size and $d$ is the average degree). While in the latest comment, Reviewer UTjD claims that:
> > > > For every epoch of training, Cluster-GCN does not need to redo clustering, and all it does is to pick the clusters--so there is no overhead for clustering at all.
> > >
> > > , from which we see that **Reviewer UTjD did not realize there is an $O(bd)$ overhead to add the between-cluster edges back for each mini-batch**. This $O(bd)$ overhead per mini-batch still leads to an $O(m)$ overhead per epoch as there are $O(n/b)$ mini-batches in an epoch, where $n$ is the number of nodes. Let us explain why Cluster-GCN has this $O(bd)$ time overhead per mini-batch in detail as follows.
> > >
> > >  - Firstly, it is stated in the original Cluster-GCN paper [1] that, **for each mini-batch**, when Cluster-GCN randomly picks the $q$ clusters, it also has to **add the edges between the chosen clusters back** to the sampled adjacency matrix. This statement is in section 3.2 on page 5 of [1] and is quoted as follows:
> > >     > When constructing a batch $B$ for an SGD update, instead of considering only one cluster, we randomly choose $q$ clusters, denoted as $t_1,\dots,t_q$ and include their nodes \{$V_{t_1}\cup\cdots\cup V_{t_q}$\} into the batch. Furthermore, **the links between the chosen clusters, \{$A_{ij} | i,j \in t_1,\dots,t_q$\}, are added back**.
> > >  - Secondly, in our experiments, we use the implementation of Cluster-GCN in the famous *PyTorch Geometric (PyG)* library [3]. We can see there are [three lines of code](https://github.com/rusty1s/pytorch_geometric/blob/e425622d6efc6832b15e9fe577710a7119d76cef/torch_geometric/data/cluster.py#L158-L160) (L158-L160) in their repository, which is responsible for this "incorporating between-cluster edges" process. **This process is not trivial** because the sparse adjacency matrice is stored in any of the COO, CSR, CSC formats in *PyG* (see [the source code](https://github.com/rusty1s/pytorch_sparse/blob/2f3ecc3eb9faad495d23252f0d04644426a78024/torch_sparse/storage.py#L16) in the auxiliary *pytorch_sparse* package). And to select all the within-cluster and between-cluster edges, we have to (1) do row slicing on the original adjacency matrix and concatenate the selected matrices ([L158](https://github.com/rusty1s/pytorch_geometric/blob/e425622d6efc6832b15e9fe577710a7119d76cef/torch_geometric/data/cluster.py#L158)), (2) do column slicing on the concatenated matrix ([L159](https://github.com/rusty1s/pytorch_geometric/blob/e425622d6efc6832b15e9fe577710a7119d76cef/torch_geometric/data/cluster.py#L159)), and (3) convert the selected part of adjacency matrix to COO format ([L160](https://github.com/rusty1s/pytorch_geometric/blob/e425622d6efc6832b15e9fe577710a7119d76cef/torch_geometric/data/cluster.py#L160)). We are required to convert among the COO, CSR, and CSC formats and the time complexity is linear in the number of non-zero elements. The number of non-zero elements in the concatenated matrix in [L158](https://github.com/rusty1s/pytorch_geometric/blob/e425622d6efc6832b15e9fe577710a7119d76cef/torch_geometric/data/cluster.py#L158) is approximately $O(bd)$. Thus the time complexity of these three lines of code, which is repeated for each mini-batch, is also $O(bd)$.
> > >  - To conclude, the total time overhead per epoch for Cluster-GCN to "add between-cluster edges back" is still $O(m)$, although it is much smaller than the time to cluster the entire graph at the beginning.
> > >
> > > ## For GraphSAINT-RW
> > > Reviewer UTjD claims the following in the new comment.
> > > > "Similar for GraphSaint, the batching time is not related to L(number of layers)."
> > >
> > > We respectfully point out that **this is a misunderstanding of the Reviewer.** We want to address again that **the mini-batch sampling time of GraphSAINT-RW is proportional to $L$, the number of layers because in the original GraphSAINT paper [2] and in our manuscript, the walk length of the random walk sampler is usually set to $L$.** This is clearly stated
> > >  -  in the original GraphSAINT paper [2] (in the "Random walk based samplers" paragraph on page 6):
> > > > "Even though it is not possible to sample a subgraph where such pairs of nodes are independently selected, **we still consider a random walk sampler with walk length $L$ as a good candidate for $L$-layer GCNs**."
> > >  - in the submitted manuscript (L280-282):
> > > > "We consider the best-performing variant, GraphSAINT-RW, **which uses $L$ steps of random walk** to induce subgraph from $b$ randomly sampled nodes."
> > >  - and in the response to Reviewer UTjD's Concern 3 above:
> > > > "GraphSAINT-RW also requires a relative expensive random walk simulation process on the large graph to sample the mini-batches, which is $O(Lbd)$ where $L$ is the number of layers of GNNs and **also the number of steps of random walks**, $b$ is the batch size, and $d$ is the average degree.
> > >
> > > To conclude, it is because that
> > > 1. the mini-batch sampling time of the random walk sampler is proportional to the walk length (i.e., number of steps of each random walk)
> > > 2. and the walk length is usually set to $L$, the number of layers of GNNs, in [2] and our paper
> > >
> > > , we say the mini-batch sampling time of GraphSAINT-RW is proportional to $L$, the number of layers.
> > >
> > > ### References:
> > >  * [1] Chiang, W. L., Liu, X., Si, S., Li, Y., Bengio, S., & Hsieh, C. J. (2019, July). Cluster-GCN: An efficient algorithm for training deep and large graph convolutional networks. In Proceedings of the 25th ACM SIGKDD International Conference on Knowledge Discovery & Data Mining (pp. 257-266).
> > >  * [2] Zeng, H., Zhou, H., Srivastava, A., Kannan, R., & Prasanna, V. (2019). Graphsaint: Graph sampling-based inductive learning method. arXiv preprint arXiv:1907.04931.
> > >  * [3] Fey, M., & Lenssen, J. E. (2019). Fast graph representation learning with PyTorch Geometric. arXiv preprint arXiv:1903.02428.

---

> > > > ### Comment · Reviewer_UTjD · 2021-09-03
> > > > **still not convinced**
> > > >
> > > > Thanks for authors' explanation. To be honest I am not fully convinced by the replies. First of all, for clusterGCN, the actually computation time for merging clustering should be even less than E, while with the time complexity to be O(E). All it does is just to merge small clusters (just one pass over the graph) into larger ones, so it should be very fast. I would suggest digging into the codes and check why it is slow. Without experimental evidence of the actually overhead time cost, I think it is hard to remove my concerns on these experiment results. Similarly for GraphSaint.

---

> > > > > ### Author Response · Authors · 2021-09-06
> > > > > **Reply to Reviewer UTjD's concern**
> > > > >
> > > > > We are grateful for Reviewer UTjD's latest response. As suggested, we have carefully checked our code of Cluster-GCN and GraphSAINT-RW (which are based on the implementations in PyG [3]), and they are sound to the best of our knowledge and efforts. One thing we did find helped speed up the baselines is to downgrade the `num_workers` hyperparameter of the data loaders from 4 to 0, which improves per epoch time from 2.22s to 1.86s for Cluster-GCN and 2.10s to 1.56s for GraphSAINT-RW, respectively. We initially set the `num_workers` to 4 for time evaluation because we thought it was optimal for a machine with a 4-core CPU and sufficient memory.
> > > > >
> > > > > Considering that reporting total training time is the common practice as in the baseline papers [1, 2], we reformulate the time statistics in the following table (as discussed above, we set `num_workers` to 0 for all experiments). Here total training time includes data loading, pre-processing, and training (forward-pass and back-propagation) but excludes validation set evaluation and model saving. We train all the models for ten epochs as it is enough for them to converge on the ogbn-arxiv dataset. There are 16 mini-batches in an epoch, and the mini-batch size is fixed at 10k.
> > > > >
> > > > > | Method                | Total training time |
> > > > > | --------------------- | ------------------- |
> > > > > | Cluster-GCN (PyG [3]) | 29.1s               |
> > > > > | GraphSAINT-RW (PyG)   | 19.2s               |
> > > > > | VQ-GNN (PyG)          | 12.8s               |
> > > > >
> > > > > As a sanity check, we further implemented the two baselines using the DGL [4] library (former baselines are on PyG [3]), and we found that DGL provides faster utility functions for Cluster-GCN and GraphSAINT-RW, which leads to reduced total time. In this regard, we believe the slow training time of the baselines is because of the less efficient implementations in PyG. Although due to the limited time of the review process, we cannot rebuild our VQ-GNN algorithm on DGL now, we see the hope to speed up our method further and will try DGL in the future. The above discussions did help us get a better understanding of the baselines and our methods, and we really appreciate that. We will add a more detailed time analysis in the future version as the Reviewer expected and opensource our code.
> > > > >
> > > > > | Method                | Total training time |
> > > > > | --------------------- | ------------------- |
> > > > > | Cluster-GCN (DGL [4]) | 7.0s                |
> > > > > | GraphSAINT-RW (DGL)   | 6.8s                |
> > > > >
> > > > > ### References:
> > > > >  * [1] Chiang, W. L., Liu, X., Si, S., Li, Y., Bengio, S., & Hsieh, C. J. (2019, July). Cluster-GCN: An efficient algorithm for training deep and large graph convolutional networks. In Proceedings of the 25th ACM SIGKDD International Conference on Knowledge Discovery & Data Mining (pp. 257-266).
> > > > >  * [2] Zeng, H., Zhou, H., Srivastava, A., Kannan, R., & Prasanna, V. (2019). Graphsaint: Graph sampling-based inductive learning method. arXiv preprint arXiv:1907.04931.
> > > > >  * [3] Fey, M., & Lenssen, J. E. (2019). Fast graph representation learning with PyTorch Geometric. arXiv preprint arXiv:1903.02428.
> > > > >  * [4] Wang, M., Zheng, D., Ye, Z., Gan, Q., Li, M., Song, X., ... & Zhang, Z. (2019). Deep graph library: A graph-centric, highly-performant package for graph neural networks. arXiv preprint arXiv:1909.01315.

---

> > > ### Author Response · Authors · 2021-09-03
> > > **We are looking forward to your reply**
> > >
> > > Dear Reviewer UTjD, thank you again for reading our long rebuttal and raising some new questions. We know that you are worried that our authors have some conceptual misunderstanding of the baselines. However, we have used sentences in the original submitted manuscript to prove that it's a writing mistake only in the rebuttal, and we did not misunderstand the baselines in the paper and experiments. We hope our response below could clarify the problem and resolve your concern. Thank you!

---

### Author Response · Authors · 2021-08-10
**Common Response 1: Experiments on Extra Benchmarks**

### 1. Reviewers' concerns:
Multiple reviewers have raised their concerns regarding our experimental results:
1. **Reviewer UTjD: larger benchmarks required.** Reviewer UTjD criticized the inadequate evaluation of our proposed VQ-GNN framework on only one dataset (i.e., *ogbn-arxiv*) for node classification and one dataset (i.e., *ogbl-collab*) for link prediction. Reviewer UTjD would like to see larger-scale experiments with node sizes in millions.
2. **Reviewer UTjD: experiments under inductive learning setting.** The two benchmarks in the manuscript (*ogbn-arxiv* and *ogbl-collab*) were transductive learning tasks. Reviewer UTjD was curious whether our proposed VQ-GNN framework can be applied under the inductive learning setting. More specifically, Reviewer UTjD would like to see the performance of VQ-GNN on some inductive learning benchmarks.
3. **Reviewer ai2z: limited experiments.** Reviewer ai2z commented that our experimental results are convincing but limited. More experimental results are demanded.
4. **Reviewer f3BS: not outstanding performance.** Reviewer f3BS argued that the generalization performance of our proposed VQ-GNN framework was not outstanding enough and did not show unique values among several solutions that provide scalability for GNNs. To be honest, more experimental results are needed to draw a conclusion on the performance of VQ-GNN.

We really appreciate the thoughtful ideas regarding the extra benchmarks from the reviewers. We agree that the main weakness of the manuscript is in the limited experiments. We hope the following newly added experiments can strengthen our experimental results and help the reviewers re-evaluate the paper.

### 2. Four extra benchmarks:
In this response, we are able to evaluate the proposed VQ-GNN framework on **four more benchmarks: *FLICKR* [1], *REDDIT* [2], *PPI* [2], and *ogbl-citation2* [3]**. We want to highlight some details of the newly added datasets to the reviewers:
1. ***ogbl-citation2* has over 2.9 million nodes.** *ogbl-citation2* is a link prediction benchmark with over 2.9 million nodes and 30.5 million edges. Experiments on this large-scale graph can demonstrate the scalability of the VQ-GNN framework. The new experiments address the concern of **Reviewer UTjD**.
2. ***PPI* is an inductive learning task.** *PPI* is a node classification benchmark under the inductive learning setting, i.e., neither attributes nor connections of test nodes are present during training. **VQ-GNN can be applied under inductive setting with only one extra step**: during the inference stage, we now need to find the codeword assignments (i.e., the nearest codeword) of the test nodes before making predictions since we have no access to the test nodes during training. Neither the learned codewords nor the GNN parameters are updated during inference.
3. ***FLICKR* and *REDDIT* has more node attributes than the *ogb* datasets, and *REDDIT* is a relatively dense graph.** Usually *ogb* datasets come with 128 or fewer node attributes. *FLICKR* and *REDDIT* have 500 and 602 features per node, respectively. The increased dimensionality of input node features may be challenging for VQ-GNN because VQ has to compress higher-dimensional vectors. Moreover, *REDDIT*'s average node degree is around 50. More memory is required for mini-batches of the same size because more messages are passed in a layer of GNN[\*].

Detailed statistics of all datasets used in the experiments are summarized in the following table (*ogbn-arxiv* and *ogbl-collab* are used in Section 6 of the manuscript). This table and more details of the datasets will be added to the Appendix.

| Dataset       | ogbn-arxiv   | ogbl-collab  | FLICKR       | REDDIT       | PPI          | ogbl-citation2 |
|---------------|--------------|--------------|--------------|--------------|--------------|----------------|
| Task          | node         | link         | node         | node         | node         | link           |
| Setting       | transductive | transductive | transductive | transductive | inductive    | transductive   |
| Label         | single       | single       | single       | single       | multiple[\^] | single         |
| # of Nodes    | 169,343      | 235,868      | 89,250       | 232,965      | 56,944       | 2,927,963      |
| # of Edges    | 1,166,243    | 1,285,465    | 449,878      | 11,606,919   | 793,632      | 30,561,187     |
| # of Features | 128          | 128          | 500          | 602          | 50           | 128            |
| # of Classes  | 40           | (2)          | 7            | 41           | 121          | (2)            |
| Label Rate    | 54.00%       | 92.00%       | 50.00%       | 65.86%       | 78.86%       | 98.01%         |

### 3. Performance comparison:
We evaluate our proposed VQ-GNN framework on the four new benchmarks, as shown in the table below. Similar to the experiments in Section 6 of the manuscript, for each dataset and if possible, we try to test VQ-GNN with three types of GNN backbones (i.e., GCN, GraphSAGE with the mean aggregator, and GAT) and compare it with NS-SAGE[\@] [2], Cluster-GCN [4], and GraphSAINT-RW [1] sampling-based baselines.

| Benchmark     | FLICKR       | FLICKR       | FLICKR       | REDDIT       | REDDIT       | REDDIT        | PPI[\#]      | PPI          | PPI          | ogbl-citation2 | ogbl-citation2 |
|---------------|--------------|--------------|--------------|--------------|--------------|---------------|--------------|--------------|--------------|----------------|---------------|
| GNN Model     | GCN          | SAGE-Mean    | GAT          | GCN          | SAGE-Mean    | GAT           | GCN          | SAGE-Mean    | GAT          | GCN            | SAGE-Mean      |
| "Full Graph"  | .5209± .0053 | .5177± .0041 | .5156± .0067 | OOM[\$]      | OOM          | OOM           | .9173± .0039 | .9358± .0046 | .9722± .0035 | OOM[\%]        | OOM            |
| NS-SAGE       | N/A          | .5165± .0077 | .5282± .0052 | N/A          | .9615± .0089 | .9426± .0043  | N/A          | .9121± .0033 | .9407± .0025 | N/A            | .7881± .0078   |
| Cluster-GCN   | .4976± .0078 | .4996± .0045 | .4907± .0107 | .9264± .0034 | .9456± .0061 | .9380± .0055  | .8852± .0066 | .8810± .0091 | .9051± .0077 | .7897± .0057   | .7097± .0089   |
| GraphSAINT-RW | .5239± .0071 | .5040± .0046 | .5163± .0062 | .9225± .0057 | .9581± .0074 | .9431± .0067  | .9110± .0057 | .9382± .0074 | .9612± .0042 | .7954± .0078   | .7858± .0061   |
| VQ-GNN (Ours) | .5315± .0031 | .5323± .0083 | .5288± .0054 | .9399± .0021 | .9449± .0024 | .9438± .0059  | .9549± .0058 | .9324± .0043 | .9737± .0033 | .7961± .0089   | .8024± .0135   |

We want to highlight some details in the table above to the reviewers:
1. **VQ-GNN outperforms the baselines by a margin on *FLICKR*.** We see our VQ-GNN framework can outperform the full-graph training and sampling-based baselines by a margin[\&] on the *FLICKR* benchmark using all the three GNN backbones. These demonstrate that VQ-GNN can show outstanding performance among the other scalability methods on some real-world benchmarks and address the concern of **Reviewer f3BS**.
2. **VQ-GNN works well under inductive setting on *PPI*.** Under the inductive setting, the test nodes' attributes are not accessible during training and thus not involved in the codebook learning process of VQ-GNN. Thus the codewords can only represent the training nodes' feature distribution. From the experiments on *PPI*, we see this restriction does not noticeably harm the performance of VQ-GNN. This may be because (1) the codebook is a coarse summarization of the distribution and thus generalizes well, and (2) the training and test feature distributions of *PPI* are very similar. Experiments on the *PPI* benchmark show that VQ-GNN can be applied to both transductive and inductive learning on graphs and address the concern of **Reviewer UTjD**.
3. **VQ-GNN shows robust performance across all six benchmarks.** The six graph datasets we have considered so far have diverse topology and statistics, and we can see there is no single scalable algorithm that consistently outperforms the others. VQ-GNN is robust as it can achieve competitive performance on nearly all benchmarks with different GNN backbones, while the baselines may suffer from performance drop in some cases.

### Remarks in this response:
* [\*] For some implementations of GNNs, e.g., the PyG library [5] this paper use, the memory usage is proportional to the number of messages passed in a layer of GNN, instead of the number of nodes in the mini-batch.
* [\^] *PPI* benchmark comes with multiple labels per node, and the evaluation metric is F1 score instead of accuracy.
* [\@] We call the neighbor sampling method in [2] NS-SAGE and the GNN model in the same paper SAGE-Mean to avoid ambiguity.
* [\#] The SOTA performance on the *PPI* benchmark is generally achieved using very wide GNNs (e.g., 5-layer GCNs with 2048 neurons per layer in [1]). Here, for our experiments on *PPI*, we use 3-layer GNNs with 256 neurons.
* [\$] "OOM" refers to "out of memory." *REDDIT* node classification benchmark comes with 602 node features, and the average node degree is around 50. We find that full-graph training of GNNs on *REDDIT* requires more than 12GB of memory, although the size of the graph is not very large.
* [\%] In [3], it is reported that the "full graph" training performance of 3-layer GCNs with 256 neurons per layer is 0.8474±0.0021, and the performance of 3-layer SAGE-Mean with 256 neurons per layer is 0.8260±0.0036. Our experiments on *ogbl-citation2* only use 128 neurons per layer. We are not sure about some baseline performance of GAT.
* [\&] There is a performance gap for most cases, except when training GAT and compared with NS-SAGE.

---

> ### Author Response · Authors · 2021-08-10
> **Common Response 2: More Experimental Analysis**
>
> ### 1. Reviewers' concerns:
> Multiple reviewers have suggested doing more experimental analysis to understand how sensitive the proposed VQ-GNN framework is to the important hyper-parameters:
> 1. **Reviewer UTjD: performance vs. the number of layers.** Reviewer UTjD was curious about how the performance varies with respect to the number of layers of GNNs. In the experiments, we always set the number of layers to 3 in the experiments to fairly compare VQ-GNN with the baselines. Now we want to check how the performance of VQ-GNN is affected by the depth of GNNs.
> 2. **Reviewer UGM2 and Reviewer ai2z: performance vs. codebook and mini-batch sizes.** Reviewer UGM2 would like to see experimental analysis on how the performance depends on the codebook and mini-batch sizes. Reviewer ai2z was expecting a similar analysis: "how the hyper-parameter selections such as reduced dimension (i.e., the codebook size) and the mini-batch sizes affect the algorithm." Codebook and mini-batch sizes are indeed two important hyper-parameters of the VQ-GNN algorithm. We also think it is interesting to see how they affect the performance.
> 3. **Reviewer ai2z: performance vs. mini-batch sampling strategies.** Reviewer ai2z would prefer to see some discussion on how the mini-batch sampling strategies affect our VQ-GNN algorithm. Our VQ-GNN does not rely on specific mini-batch sampling strategies, but the performance could be affected by how you sample the mini-batches. In addition to just randomly sampling nodes, we consider two other sampling strategies: randomly sampling edges and sampling using random walks as GraphSAINT-RW [1].
>
> We appreciate the thoughtful ideas on the experimental analysis and have closely followed the suggestions. We hope the reviewers will find the discussions below helpful.
>
> ### 2. Experimental analysis:
> We conducted several new experiments on the *ogbn-arxiv* dataset with GCN backbone as follows. Please note that if not mentioned otherwise, the hyper-parameter setups will follow the corresponding model in the manuscript and are described in Appendix F.
> 1. **Performance vs. the number of layers:** We vary the number of layers of GCN backbone from one to five, and the performance of VQ-GNN on *ogbn-arxiv* is shown in the following table. We see that a two-layer GCN is sufficient to achieve good performance on *ogbn-arxiv* while using only one layer harms the performance. Stacking more layers will not bring any performance gain.
>
> | # of Layers | 1      | 2      | 3      | 4      | 5      |
> | ----------- | ------ | ------ | ------ | -----  | ------ |
> | Accuracy    | 0.6599 | 0.7006 | 0.7055 | 0.7080 | 0.7037 |
>
> 2. **Performance vs. codebook size:** We vary the codebook size from 64 to 4096 and fix the mini-batch size at 40K. Performance is shown as follows, where we can see the performance is not sensitive to the codebook size. Setting the codebook size to 64 is enough to achieve relatively good performance on *ogbn-arxiv*. This may indicate that the node feature distribution of *ogbn-arxiv* is sparse, which is reasonable as the 128-dimensional node features are obtained by averaging the embedding of words in the titles and abstracts of arXiv papers [3].
>
> | Codebook Size | 64     | 256    | 1024   | 4096   |
> | ------------- | ------ | ------ | ------ | ------ |
> | Accuracy      | 0.6950 | 0.7011 | 0.7030 | 0.7049 |
>
> 3. **Performance vs. mini-batch size:** We vary the mini-batch size from 10K to 80K, and the performance is shown as follows. We see that smaller mini-batch sizes can slightly lower the overall performance. When the mini-batch size is smaller, more messages come from out-of-mini-batch nodes (see Figure 1 of the manuscript), and they are approximated by the VQ codewords. This increased number of approximated messages can harm the performance. However, generally speaking, the performance is not sensitive to the mini-batch size as long as it is not small.
>
> | Mini-batch Size | 10K    | 20K    | 40K    | 60K    | 80K    |
> | --------------- | ------ | ------ | ------ | ------ | -----  |
> | Accuracy        | 0.6843 | 0.6920 | 0.7011 | 0.7055 | 0.7061 |
>
> 4. **Performance vs. mini-batch sampling strategy:** We compare three different sampling strategies: (1) randomly sampling nodes, (2) randomly sampling edges, and (3) sampling using random walks as in GraphSAINT-RW [1]. As shown in the following table, we did not observe an obvious difference in the performance.
>
> | Mini-batch Sampling Techniques | Sampling Nodes | Sampling Edges | Sampling using Random Walks|
> | ------------------------------ | -------------- | -------------- | -------------------------- |
> | Accuracy                       | 0.7020         | 0.7034         | 0.7023                     |
>
>
> ### References in Common Response 1 and 2:
> * [1] Zeng, H., Zhou, H., Srivastava, A., Kannan, R., & Prasanna, V. (2019). Graphsaint: Graph sampling-based inductive learning method. arXiv preprint arXiv:1907.04931.
> * [2] Hamilton, W. L., Ying, R., & Leskovec, J. (2017, December). Inductive representation learning on large graphs. In Proceedings of the 31st International Conference on Neural Information Processing Systems (pp. 1025-1035).
> * [3] Hu, W., Fey, M., Zitnik, M., Dong, Y., Ren, H., Liu, B., Catasta, M., & Leskovec, J. (2020). Open Graph Benchmark: Datasets for Machine Learning on Graphs. ArXiv, abs/2005.00687.
> * [4] Chiang, W. L., Liu, X., Si, S., Li, Y., Bengio, S., & Hsieh, C. J. (2019, July). Cluster-GCN: An efficient algorithm for training deep and large graph convolutional networks. In Proceedings of the 25th ACM SIGKDD International Conference on Knowledge Discovery & Data Mining (pp. 257-266).
> * [5] Fey, M., & Lenssen, J. E. (2019). Fast graph representation learning with PyTorch Geometric. arXiv preprint arXiv:1903.02428.

---

### Author Response · Authors · 2021-08-27
**Follow-up and a kind reminder**

We want to thank all the reviewers, again, for the constructive comments and thoughtful reviews. As a follow-up on our rebuttal, we would like to kindly remind the reviewers that the discussion period is ending soon. We hope to use this open response period to discuss the paper, answer additional questions, and ultimately improve the quality of our submission. Have you gotten a chance to read our responses below, which attempt to address all of your concerns? We want to make sure that the reviewers found our responses coherent and convincing. And we would be more than happy to provide more information or clarification, should it be necessary.

---

> ### Comment · Reviewer_f3BS · 2021-08-29
> **Thank you for your feedbacks!**
>
> Authors,
>
> Thank you for your detailed feedbacks.
> Feedbacks to my questions are satisfactory, solving some of my unclear misunderstanding issues.
> I will update my review and evaluations afterwards, also referring other reviewers' opinions.
>
> Thank you again!

---

> ### Comment · Reviewer_UGM2 · 2021-08-31
> **Detailed rebuttal**
>
> Thanks to the authors for the detailed rebuttal. Personally my questions / comments were addressed in a satisfactory way, and I'm happy to maintain my score of 7. If any of the reviewers have not updated their scores in response to the rebuttal, I'd recommend that they take a look, as the authors have worked hard to answer questions / comments.

---

### Decision · Program_Chairs · 2021-09-27

**Decision:**

Accept (Poster)

**Comment:**

3 of 4 ratings were "accept". The concerns from the reviewer who gave the lowest rating (a 5) were not that clear to me through the extended  conversation with the authors, and I'm not swayed by their concerns.

The authors did extensive further experiments in response to the feedback, which reinforced their results.

Overall the believe the paper is a borderline-to-clear accept, given the novelty and performance.